# Actomyosin-dependent dynamic spatial patterns of cytoskeletal components drive mesoscale podosome organization

Marjolein B.M. Meddens[1,†], Elvis Pandzic[2,†], Johan A. Slotman[3], Dominique Guillet[2], Ben Joosten[4], Svenja Mennens[4], Laurent M. Paardekooper[1], Adriaan B. Houtsmuller[3], Koen van den Dries[4], Paul W. Wiseman[2] & Alessandra Cambi[4]

Podosomes are cytoskeletal structures crucial for cell protrusion and matrix remodelling in osteoclasts, activated endothelial cells, macrophages and dendritic cells. In these cells, hundreds of podosomes are spatially organized in diversely shaped clusters. Although we and others established individual podosomes as micron-sized mechanosensing protrusive units, the exact scope and spatiotemporal organization of podosome clustering remain elusive. By integrating a newly developed extension of Spatiotemporal Image Correlation Spectroscopy with novel image analysis, we demonstrate that F-actin, vinculin and talin exhibit directional and correlated flow patterns throughout podosome clusters. Pattern formation and magnitude depend on the cluster actomyosin machinery. Indeed, nanoscopy reveals myosin IIA-decorated actin filaments interconnecting multiple proximal podosomes. Extending well-beyond podosome nearest neighbours, the actomyosin-dependent dynamic spatial patterns reveal a previously unappreciated mesoscale connectivity throughout the podosome clusters. This directional transport and continuous redistribution of podosome components provides a mechanistic explanation of how podosome clusters function as coordinated mechanosensory area.

[1] Department of Tumor Immunology, Radboud Institute for Molecular Life Sciences, Radboud University Medical Center, Geert Grooteplein Zuid 26-28, 6525 GA Nijmegen, The Netherlands. [2] Departments of Physics and Chemistry, McGill University Otto Maass (OM) Chemistry Building, 801 Sherbrooke Street West, Montreal, Quebec, Canada H3A 0B8. [3] Department of Pathology, Josephine Nefkens Institute, Erasmus MC, 3000 CA Rotterdam, The Netherlands. [4] Department of Cell Biology, Radboud Institute for Molecular Life Sciences, Radboud University Medical Center, Geert Grooteplein Zuid 26-28, 6525 GA Nijmegen, The Netherlands. † Present addresses: Department of Physics and Astronomy and Department of Pathology, University of New Mexico, 1919 Lomas Blvd. NE, Albuquerque, New Mexico 87131, USA (M.B.M.M.); Biomedical Imaging Facility, Lowy Cancer Research Centre (C25), Kensington, UNSW Australia, Sydney, New South Wales 2025, Australia (E.P.). Correspondence and requests for materials should be addressed to P.W.W. (email: paul.wiseman@mcgill.ca) or to A.C. (email: alessandra.cambi@radboudumc.nl).

Understanding how cells mechanically explore their surroundings is an area of intense investigation[1]. Depending on cell type and substrate physicochemical properties, different cellular structures and molecular mechanisms have been investigated that allow cells to anchor to, remodel and migrate through the extracellular matrix[2–6]. Common to all these processes is the capacity of actin filaments to organize in different assemblies by specifically interacting with actin-binding proteins such as talin, vinculin and non-muscle myosin IIA. Ranging from adhesive actin waves, to filopodia, focal adhesions (FAs) and podosomes, these different actin-driven multimolecular assemblies exhibit highly specific architectures and dynamic behaviour[7–10]. This diversity is particularly intriguing considering that these actin structures share many common molecular components.

Podosomes facilitate immune cell migration through barriers such as the vascular endothelium and basal membranes[11–14], sense substrate mechanical properties and topographical cues[15,16], and are involved in local bone matrix remodelling by osteoclasts[17,18]. In dendritic cells (DCs), specialized immune cells that slowly migrate through peripheral tissues patrolling for foreign antigens[19,20], podosomes most likely facilitate the passage through tissue barriers with varying architectures[11], similarly to what is reported for macrophages undergoing mesenchymal migration in three-dimension (3D)[21].

Podosomes consist of a dense core of filamentous actin (F-actin) and actin associated proteins such as Arp2/3 and cortactin that is surrounded by integrins and cytoskeletal adaptors like talin and vinculin that likely link the integrins to the actin core[22]. Podosomes are very dynamic, and continuous turnover of the actin core and several adaptor proteins has been described[18,23,24]. Furthermore, individual podosomes exhibit stiffness oscillations and density fluctuations of actin and mechanosensitive components such as vinculin and zyxin that depend on myosin IIA contractility[23,25].

By scanning electron microscopy in osteoclasts and super resolution stochastic optical reconstruction microscopy in primary human antigen-presenting DCs we, and others, revealed that neighbouring podosomes are interconnected by actin fibers, which radiate from the cores and are collectively termed the 'actin network'[9,26]. More recently, time-lapse protrusion force microscopy (PFM) measurements in living macrophages demonstrated a local short-range spatial synchrony between podosome first neighbours both at the level of actin dynamics and the exertion of protrusive forces[27]. Together, these data support the idea of podosome clusters being well-defined compartmentalized zones in DCs[28] and suggest the existence of a mesoscale control of podosome dynamics. It remains, however, elusive how such a mesoscale organization is orchestrated at the molecular level and whether the actin network contributes to this organization.

Here we hypothesized that mesoscale dynamics exist that regulate movement and redistribution of structural components among podosomes to facilitate substrate sensing and efficient transmission of mechanical stimuli throughout the podosome cluster, possibly coordinating the cell's adhesive and protrusive activity.

By integrating nanoscopy of the podosome cluster actomyosin machinery with measurements of the spatiotemporal molecular dynamics of podosome cytoskeletal components obtained with an extension of Spatiotemporal Image Correlation Spectroscopy (STICS) for short time windows, we here investigate the dynamic and structural mechanisms that occur at podosome clusters in human DCs. Our findings show a previously unappreciated mesoscale coordination of spatiotemporal dynamics of F-actin and actin binding proteins, which depends on actin polymerization and myosin IIA contractility and may act as a platform for a continuous feedback loop for the cell's local protrusive activity, as required in front-rear polarity, migration and substrate degradation. Finally, our results highlight the podosome cluster as an example of structured intracellular space, where self-assembling dynamic spatial patterns of podosome components occur within the cluster area, guaranteeing the maintenance of the podosome cluster integrity and coordination over time.

## Results

**Podosome cluster reorganization on various substrates.** Podosomes are formed at the ventral membrane of immature DCs adhering to an underlying substrate and are typically arranged in large but well-defined clusters (Supplementary Fig. 1a,b). We demonstrated before that podosomes align along edges of topographical patterns[16], but how podosome clusters dynamically rearrange the individual units to facilitate cell-substrate interactions is poorly understood. We therefore regularly monitored immature human primary DCs transfected with LifeAct-GFP, a genetically encoded F-actin binding probe[29], undergoing random migration for several hours on glass coverslips. The podosome clusters exhibit impressive spatial rearrangements over time that seem to guide the DCs during substrate exploration (Fig. 1a) or follow topographical cues on patterned substrates (Fig. 1b). In the snapshots corresponding to these movies, we colour-coded the podosomes to highlight the sequential displacement of these structures over time as the cell moves on the glass support (Supplementary Movies 1 and 2, Supplementary Fig. 1a,b). The coordination shown by the podosome clusters in both spatial arrangements is supported by images of actin and vinculin labelled in fixed DCs that were analysed by super-resolution microscopy using structured illumination microscopy (SIM) (Supplementary Fig. 1c,d). In DCs adhering to a flat glass coverslip (Supplementary Fig. 1c), podosome clusters exhibit numerous actin fibers interconnecting neighbouring podosomes, which are also preserved, although differently arranged, when podosome clusters are allowed to follow linear substrate topographical cues (Supplementary Fig. 1d).

These observations indicate the podosome cluster plasticity in terms of spatial arrangements that are used by the DCs during substrate exploration and highlight the podosome cluster as functionally coordinated unit.

**Spatiotemporal cytoskeletal patterns at podosome clusters.** We investigated whether podosome interactivity depends on molecular transport between individual podosomes. First, we studied the existence of correlated oscillation patterns between podosomes within the same cluster by performing time lapse confocal microscopy of human primary DCs transfected with LifeAct-RFP or vinculin-GFP and measured the fluorescence intensity of individual podosomes (Fig. 1c,d; Supplementary Movies 3 and 4). By plotting the fluorescence intensity traces of neighbouring and distant podosome pairs, we observed that distant pairs show no correlated fluctuations, while overlapping patterns were observed in the intensity traces of neighbouring podosomes for both vinculin-GFP and LifeAct-RFP (Fig. 1c,d), confirming the existence of short-distance molecular communication between podosome first neighbours in the same cluster.

Next, we performed kymograph analysis of LifeAct-RFP and vinculin-GFP time series in small areas within the podosome clusters (Fig. 1e,f; Supplementary Movies 3 and 4) demonstrating that LifeAct-RFP and vinculin-GFP show traveling wave-like movements within the podosome cluster that suggest a regulated

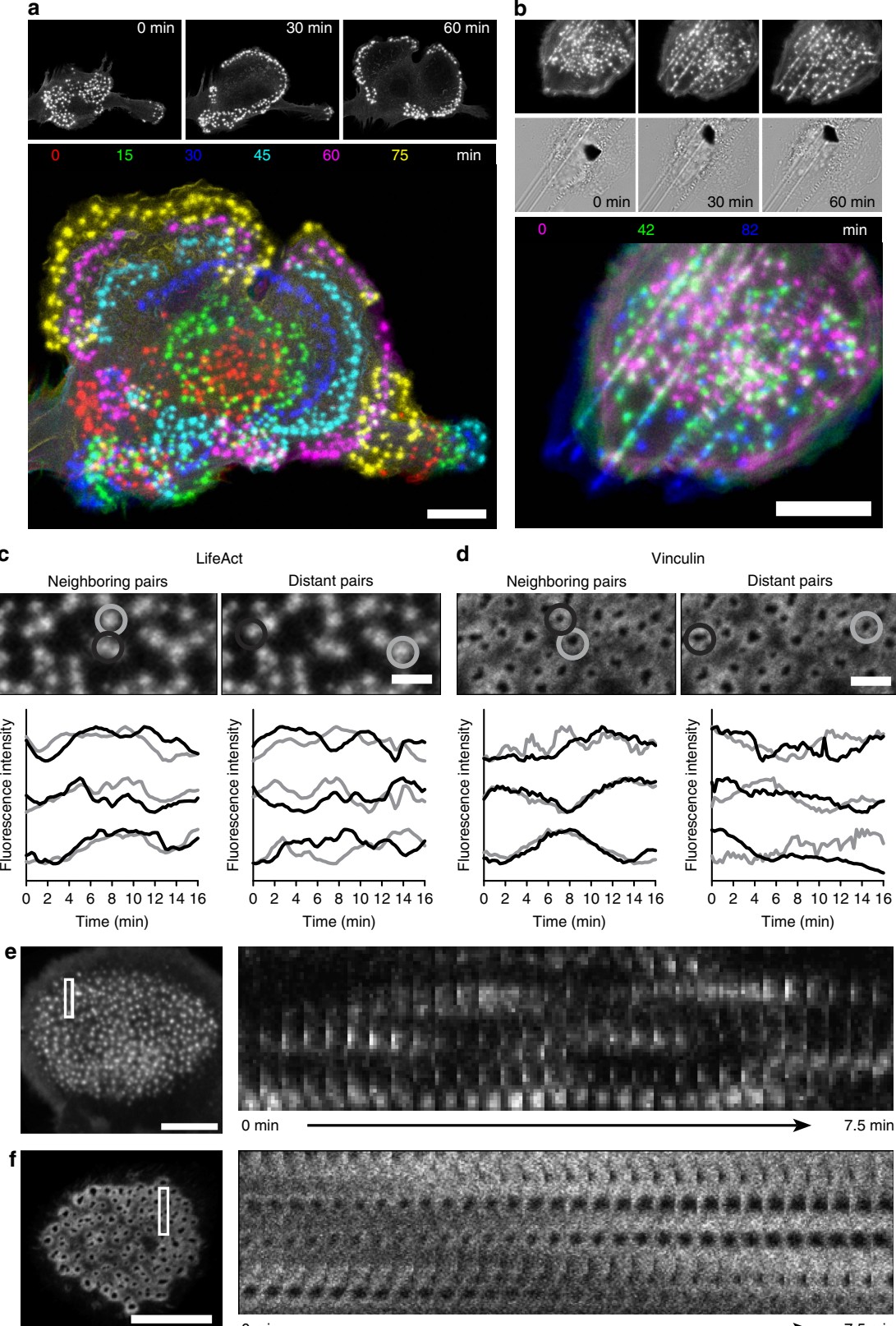

**Figure 1 | Spatiotemporal reorganization and coordinated fluctuations of podosome clusters during DC substrate probing.** DCs were transfected with LifeAct-GFP and allowed to adhere on a flat glass coverslip (**a**) or a coverslip that was scratched with sand paper (see Methods section) (**b**) and their random slow migration was monitored for several hours at 37 °C by fluorescence microscopy with 3 min frame intervals. Corresponding snapshots at specific times were colour-coded and overlaid. Both on glass and scratches >10 cells from >3 experiments were imaged and one representative movie is shown. DCs were transfected with LifeAct-RFP (**c,e**) or vinculin-GFP (**d,f**) and seeded in a glass bottom dish. Imaging was performed at a confocal microscope with 15 s frame intervals. Shown are snapshots of the movies and plots of the RFP/GFP fluctuations over time of three representative neighbouring and distant podosome pairs per condition (grey and black lines). Data are normalized to the maximum intensity. Kymographs (**e,f**) are shown of the indicated rectangular areas of the LifeAct-RFP and the vinculin-GFP movies. Scale bars 10 μm (**a,b,e,f**) and 2 μm (**c,d**).

lateral displacement of actin and vinculin. To directly visualize this displacement, we performed time lapse confocal microscopy of vinculin tagged with tdEos, a green to red photo-switchable protein, in podosome clusters. Ultraviolet-laser exposure of a specific region within the cluster was used to photo-switch the dye, and the redistribution of photo-switched tdEos-vinculin molecules was monitored over time (Fig. 2a, Supplementary Movie 5). First, we quantified the radial displacement of vinculin by monitoring the red to green intensity ratio in annular regions of interest (ROIs) around the photo-switching area (Fig. 2b). Immediately after photo-switching, most photo-switched vinculin was found in the two central annular ROIs and continued to spread outwards over time until after 30–40 s the ratio was similar for all ROIs, indicating that the green and red variants became again randomly distributed. To determine the existence of spatial patterns as observed by the kymographs above, we measured the red/green intensity ratio in radial slices of similar angular spans distributed around the photo-switching area (Fig. 2c). The intensity ratio traces show that photo-switched vinculin redistributed more quickly towards certain sectors (top right ROIs) with respect to others (bottom right ROIs), despite the presence of a similar number of podosomes in each slice. This confirms the existence of a clear directionality in the vinculin redistribution within the podosome cluster.

Collectively these results indicate that within and throughout the podosome cluster, actin and vinculin are constantly redistributed in a spatially regulated manner, suggesting that coordinated dynamics of these components might contribute to the collective behaviour of the podosome cluster.

**Directional flows of podosome components**. To quantitatively verify whether spatial patterns of cytoskeletal components of podosome clusters exist, we applied STICS analysis on microscopy image time series of fluorescently labelled podosome components (Supplementary Fig. 2). STICS has previously been used to map directional flow dynamics of focal adhesion and cytoskeletal components in cultured cells, but was only applied over longer time windows[30]. The waves of fluorescence intensity observed in podosome clusters appeared to constantly change direction and magnitude over time. Therefore, we applied a sliding short time window version of the STICS analysis using a time window of interest (TOI) of 10 image frames with the TOI window iterated sequentially by a single frame shift through the image series for each STICS analysis (Supplementary Fig. 2e). This analysis yielded time evolving vector maps of transport, which enabled us to study the temporal evolution of molecule dynamics within the podosome clusters (Supplementary Fig. 2f). A more detailed explanation of the time window STICS (twSTICS) as applied here can be found in the Supplementary Methods and Supplementary Fig. 2.

To study spatiotemporal dynamics of F-actin in podosome clusters we performed time lapse confocal microscopy of LifeAct-RFP and analysed the resulting time series with twSTICS.

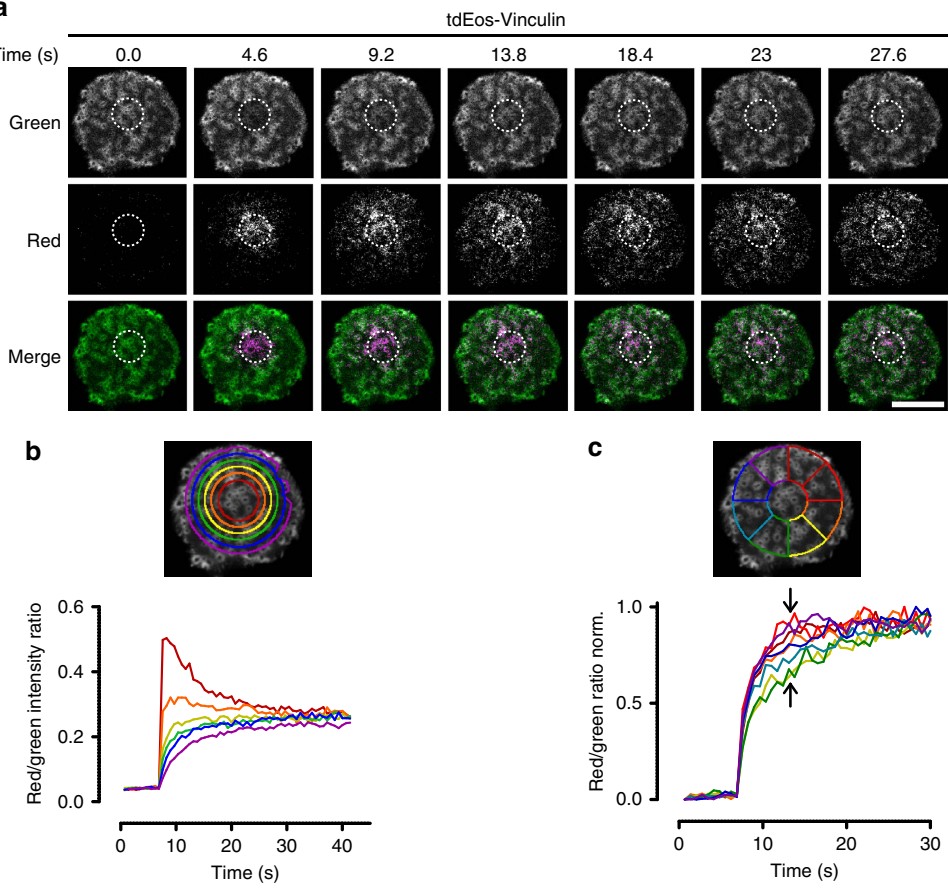

**Figure 2 | Directional redistribution of vinculin within a podosome cluster.** (**a**) DCs were transfected with tdEos-Vinculin and seeded in a glass bottom dish. Imaging of the green (non-photoswitched) and red (photoswitched) variants was performed by confocal microscopy. Photoswitching was induced by 405 nm laser short exposure in the circular area indicated by the dotted line. Redistribution of photoswitched proteins was measured in circular regions (**b**) and slices (**c**) and plotted as red to green ratio for each region (**b**). In **c** the red to green ratio was normalized to the maximum value. The colour of the plot lines corresponds with the regions indicated in the images above the plot. Scale bar, 10 µm.

Figure 3a shows original snapshots and the corresponding vector maps overlaid onto the middle image in a TOI from Supplementary Movie 6. The vectors indicate direction of flow, while the size and colour scale indicate the flow magnitude. To confirm that the on/off kinetics of the LifeAct-GFP probe did not influence measurements of filamentous actin flow, we performed fluorescence recovery after photobleaching (FRAP) analysis in podosomes, comparing actin-GFP and LifeAct-GFP (Supplementary Fig. 3a). The recovery curve of LifeAct-GFP shows full recovery within 10 s, which is faster than our STICS imaging rate (15 s per frame), thus excluding that LifeAct on/off kinetics contribute to F-actin dynamics as measured by twSTICS. We also note that the observed flow is not due to diffusion of LifeAct-RFP monomers since protein diffusion in the cytoplasm is very rapid and with a frame time of 15 s and a TOI size of 10 frames we do not detect fast diffusion which decorrelates on a shorter timescale. Moreover, we compared twSTICS results of LifeAct-GFP and actin-GFP and found no observable differences (Supplementary Fig. 3b–h and Supplementary Movie 7), indicating the suitability of LifeAct as probe for actin dynamics in the twSTICS analysis over the time scales studied here. LifeAct-RFP vector maps specifically show clear actin flow patterns (that is, groups of correlated vectors) in the podosome cluster and at the cell edges, but not in other parts of the cell where no LifeAct signal was detected (Fig. 3a). To avoid confusion, we find opportune to emphasize that with 'flow' we do not mean the actin polymerization flow rate but rather the directionality of the fluorescent features analysed by twSTICS. Moreover, it should be pointed out that actin polymerization flow rate is much faster than the velocity values obtained here.

To investigate flows of cytoskeletal adaptor proteins, we analysed DCs expressing vinculin-GFP using time lapse confocal microscopy and subsequent twSTICS analysis (Fig. 3b and Supplementary Movie 8). The resulting vector map showed that vinculin exhibits clear flow patterns throughout the podosome cluster. The direction varies in different regions of the cluster, and similarly to LifeAct-RFP, the vector maps show flow only in the podosome cluster and at the cell edges. Analysis of consecutive TOIs shows that both direction and magnitude of flows change over time, indicating a highly dynamic transport of both F-actin and vinculin throughout the podosome cluster (Fig. 3a,b; Supplementary Movies 6 and 8). Careful analysis of the vector maps reveals that the vectors appear in clear patterns throughout the podosome cluster, with direction and magnitude being correlated between neighbouring vectors and up to a distance covering multiple proximal podosomes, suggesting coordination of actin and vinculin flow at a level that goes beyond that of individual podosomes. To quantify this degree of spatiotemporal correlation within the vector maps in a cluster, we calculated the pair vector correlation (PVC), which provides heat maps indicating the degree of correlation between vectors at each spatiotemporal lag (Fig. 3c). Each point in the heat map corresponds to a specific spatiotemporal lag, and the colour scale reports the mean value of the dot product between all vector pairs from the twSTICS vector maps that are separated by that distance/time. We calculated the PVC for LifeAct and vinculin (Fig. 3d,e). These plots clearly show that there is correlation between vectors up to a temporal and spatial scale larger than what can be expected from oversampling (that is, 1.6 μm).

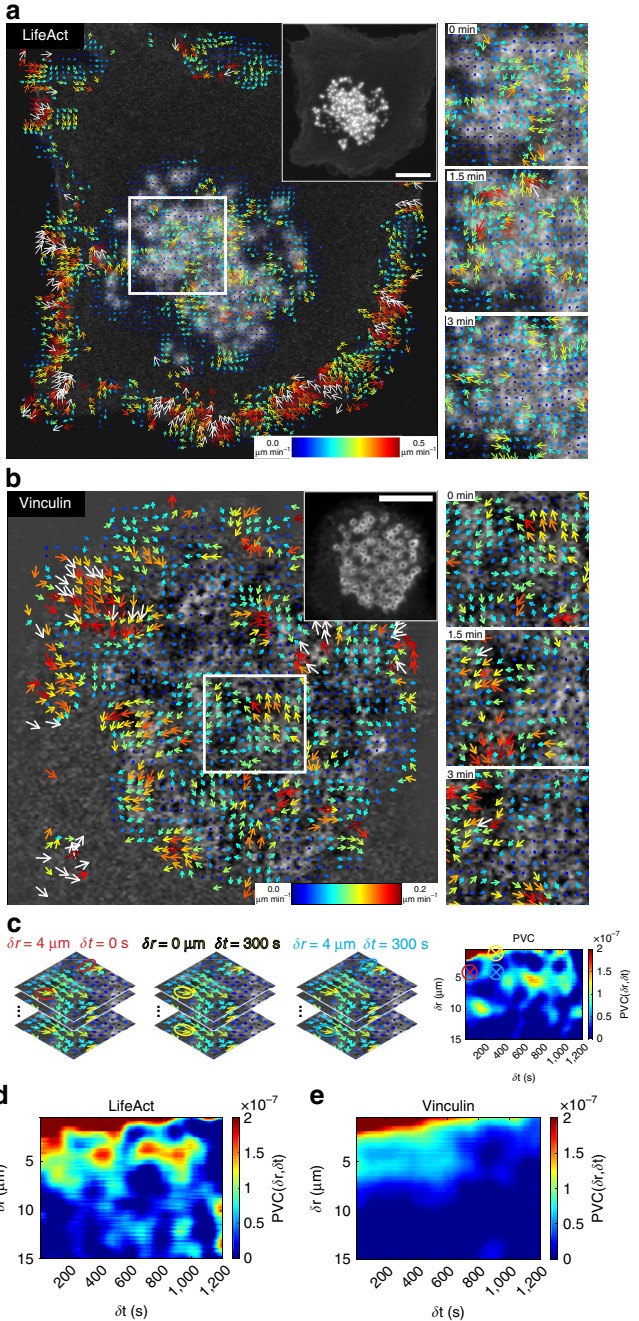

**Figure 3 | Time window STICS shows actin and vinculin flux in podosome clusters.** DCs were transfected with LifeAct-GFP (**a**) or vinculin-GFP (**b**) and seeded in a glass bottom dish. Imaging was performed at a confocal microscope with 15 s frame intervals. Time series (100 frames) were subjected to twSTICS analysis. Results are shown as vector maps in which the arrows indicate direction of flow and both the size and colour coding are representative of the flow magnitude. Vector maps are plotted onto the immobile filtered version of the images. A 10 frame moving average corresponding to the twSTICS window is shown in the upper right inset. On the right zoomed in images of the boxed region are shown for three different STICS time windows. (**c**) Explanation of Pair Vector Correlation (PVC) analysis. In PVC the spatial and temporal scales over which flow is correlated within a podosome cluster is calculated by the dot product between vectors separated in space and time. Every position in a PVC plot (right panel) is the average of all vectors at the same spatial ($\delta r$) and temporal ($\delta t$) lag. The three panels on the left denote examples of vectors at spatiotemporal lags of $\delta r = 4$ $\delta t = 0$, $\delta r = 0$ $\delta t = 300$ and $\delta r = 4$ $\delta t = 300$ and corresponding positions are denoted by the coloured circles in the PVC plot. (**d,e**) PVC plots corresponding to the movies in **a,b** show the spatial and temporal scales of vector correlation of the twSTICS analysis. Scale bars, 10 μm.

For LifeAct robust correlation can be seen up to 8 μm and for vinculin up to 6 μm, indicating that there is coordination between podosomes at spatial scales involving multiple podosomes and at temporal scales up to 15 min. These values also suggest that the spatiotemporal correlation between the vectors goes beyond that expected for first neighbour podosomes only, but rather extends further across the podosome cluster. To validate the PVC method, we separately analysed the LifeAct-GFP twSTICS vectors obtained inside and outside podosome clusters (Supplementary Fig. 4a–c). The PVC heat map for twSTICS vectors obtained inside the podosome cluster shows a similar pattern as observed earlier for LifeAct (Fig. 3d), whereas the PVC heat map for the vectors outside the podosome cluster (i.e., at the cell edge) shows larger spatiotemporal scales of correlation (Supplementary Fig. 4c), as expected from the retrograde actin flow at the cell periphery. This retrograde flow does not change direction over time, which results in positive correlation in PVC for all spatial and temporal lags. To further validate the PVC method, we randomized vector angles and magnitudes and show that this results in a PVC plot without any sign of correlation (Supplementary Fig. 4d), confirming the specificity of the positive PVC signal for vectors within podosome clusters.

To determine whether coordinated dynamics of podosome components also occur during *de novo* cluster formation, adherent DCs transfected with LifeAct-GFP were first treated with the microtubule disrupting drug nocodazole to induce global podosome dissolution followed by thorough washout of the drug. This allows us to precisely monitor the initial steps of podosome reassembly and subsequent cluster formation by time-lapse confocal microscopy[31] (Supplementary Movie 9). The time series of cluster reassembly (Fig. 4a) as well as the corresponding kymograph analysis (Fig. 4b) showed that the formation of the podosome cluster is initiated from a few podosomes in the centre of the DC ventral membrane and spreads out mainly radially within a few minutes. TwSTICS analysis of the image series demonstrated the appearance of radiating actin traveling waves during podosome cluster formation. In fact, on reassembly of the cluster a clear directionality in the F-actin flow can be seen pointing outward from the cluster centre (Fig. 4c) as also quantified in the PVC plot (Fig. 4d). This supports the idea that podosome cluster formation and expansion are at least partially achieved by directional growth of actin filaments and that structural components are directionally transported from existing towards newly formed podosomes.

The most pronounced example of coordinated dynamics is the displacement of a podosome cluster as a whole, for example during cell movement. In Supplementary Movie 10 Supplementary Fig. 4e we report an example of the lateral movement of a whole podosome cluster in DCs transfected with vinculin-mCherry, seeded onto coverslips and imaged by time-lapse confocal microscopy. In the time trace, the cluster movement is indicated by the colour-coded outlines, which show the global translation of the podosome cluster from the bottom right to the upper left part of the image (Fig. 4e). The corresponding kymograph shows that during this process, the podosomes do not move individually towards the direction of the cluster, but the cluster movement is rather achieved by podosome formation at the front and dissolution at the rear (Fig. 4f), implicating some level of coordination throughout the cluster. TwSTICS analysis of this image series showed consistent flow of vinculin pointing exactly in the direction of the overall cluster movement (Fig. 4g). This shows that even though individual podosomes do not actively move in the direction of the cluster displacement, their components follow the direction of the movement. Subsequent PVC clearly reveals that in a moving cluster the spatiotemporal scale of coordination of flow is much larger than in a static cluster (Fig. 4h). This is most pronounced spatially, indicating that, depending on the motility of the cell and the podosome cluster, shorter or longer length scale coordination of dynamics can occur.

Altogether, these results demonstrate the existence of directional flow of podosome components that accompanies spontaneous events such as formation and displacement of podosome clusters and travels throughout the podosome clusters at spatial scales spanning multiple podosomes.

**Podosome clusters respond to topographical cues.** To determine how the actin traveling waves of podosome clusters dynamically respond to sudden changes in substrate topography, we performed a series of experiments using patterned glass coverslips.

We transfected adherent DCs with LifeAct-GFP, allowed them to adhere onto scratched glass coverslips and imaged existing podosome clusters by time lapse confocal microscopy with subsequent twSTICS analysis (Fig. 5a–e and Supplementary Movie 11). Notably, podosome clusters exhibit strong collective movement along the topographical cues (Fig. 5a,b and Supplementary Movie 11), suggesting that the presence of the linear scratches provides directionality to the entire cluster. In addition, podosomes formed on the scratches display clear protruding activity that strongly correlates with the protruding activity of proximal podosomes formed on the flat surface (Fig. 5c), indicating that sensing irregular topography does not disturb the overall inter-podosome structural communication. This is supported by the observation that their overall composition on the scratches looks similar to that on flat glass surfaces (Supplementary Fig. 8b).

Interestingly, twSTICS and subsequent PVC analysis of these movies revealed how the podosome cluster globally responds to the topographical cues encountered. The presence of the scratches clearly promotes directionality of cluster movement as demonstrated by a highly increased spatiotemporal correlation among all the vectors within a cluster (Fig. 5d,e and Supplementary Movie 11).

Next we investigated whether cluster formation is affected by the presence of topographical cues. For this, DCs transfected with LifeAct-GFP and adherent to a patterned surface were first treated with nocodazole to induce global podosome dissolution followed by thorough washout of the drug to allow podosome reformation. In this way, we precisely monitored the initial steps of podosome reassembly and subsequent cluster formation by time-lapse confocal microscopy[31] (Supplementary Movie 12). The time series of cluster reassembly (Fig. 5f) as well as the corresponding kymograph analysis (Fig. 5g) showed that the formation of the podosome cluster is highly influenced by the presence of the scratches. More specifically, the scratches seem to act as nucleation sites for podosome reformation, as their presence apparently accelerate spontaneous *de novo* podosome assembly after nocodazole addition and washout. TwSTICS and subsequent PVC analysis demonstrate a high directionality in podosome reassembly in the proximity of a scratch supported by a very strong spatiotemporal correlation among all the vectors within a cluster (Fig. 5h–i).

This supports the idea that podosome clusters redistribute biochemical and structural information depending on the mechanical stimuli encountered while probing the substrate.

**Correlated traveling waves of F-actin, vinculin and talin.** Vinculin interacts with F-actin and couples F-actin flow to adhesions[32,33]. We previously found actin and vinculin to exhibit

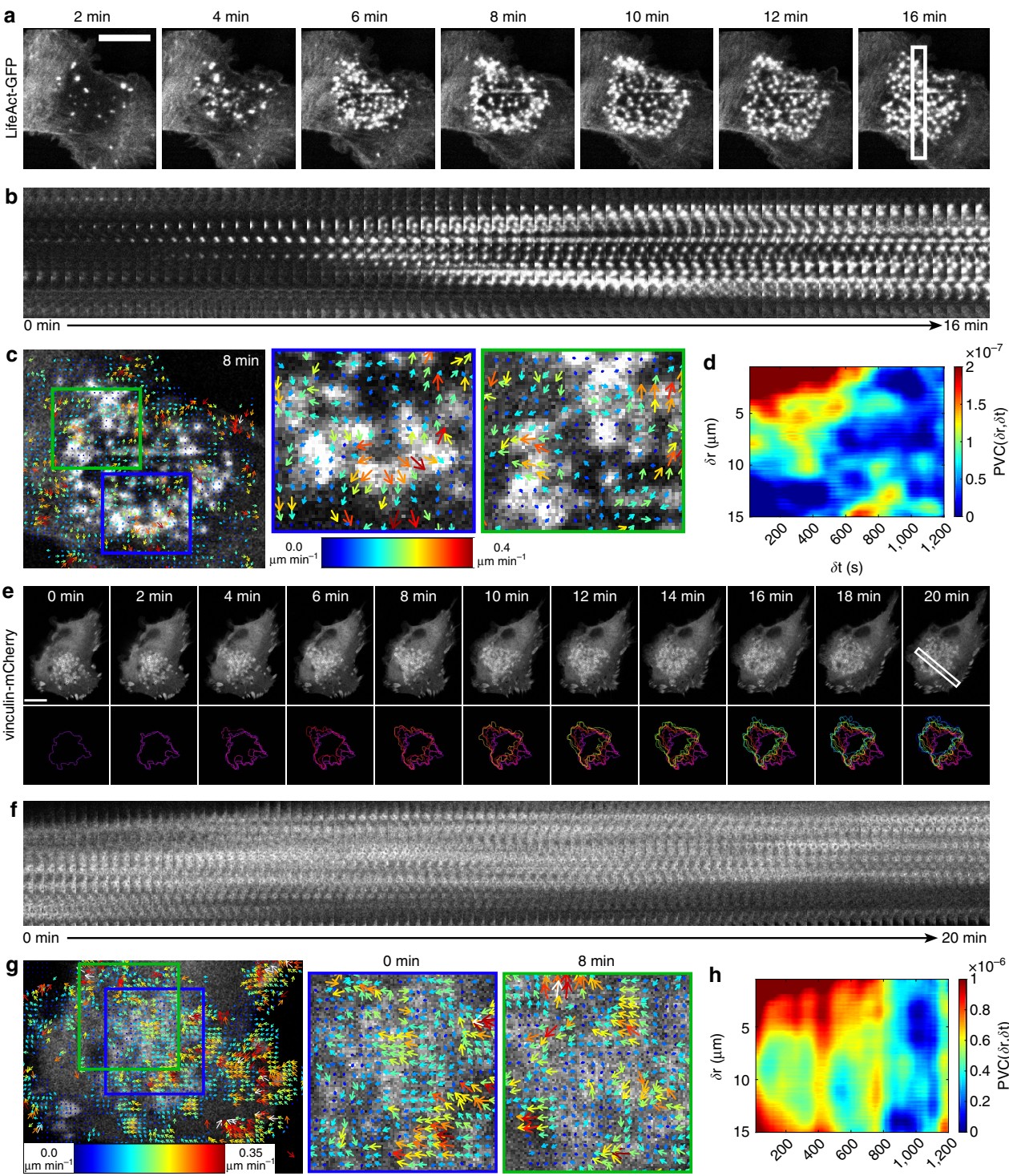

**Figure 4 | Podosome clusters show large scale flux patterns during cluster formation and movement.** (**a**–**d**) DCs were transfected with LifeAct-GFP and seeded in a glass bottom dish. Imaging was performed at a confocal microscope with 15 s frame intervals. Podosomes dissolution was induced by addition of 2–10 μM nocodazole, which was washed away when the dissolution was complete (not shown). The snapshots (**a**) and kymograph (**b**) show the part of the movie after nocodazole washout, when the podosomes reappear. The time series (300 frames) was split into 3 series of 100 frames, which were subjected to twSTICS analysis. Image and corresponding vector map plotted onto the immobile filtered version of the image are shown at 8 min after the start of cluster reformation (**c**). The arrows indicate direction of flow and both the size and colour coding are representative of the flow magnitude. The PVC plot (**d**) shows the spatial and temporal scales of vector correlation during cluster reformation on nocodazole washout. (**e**–**h**) DCs were transfected with vinculin-GFP and seeded in a glass bottom dish. Imaging was performed at a confocal microscope with 15 s frame intervals. Snapshots with corresponding images of colour coded outlines of the podosome cluster (**e**) and kymograph (**f**) show podosome cluster lateral movement. The time series was subjected to twSTICS analysis. Image and corresponding vector map plotted onto the immobile filtered version of the image are shown in **g**. The arrows in the vector maps indicate direction of flow and both the size and colour coding are representative of the flow magnitude. PVC plot (**h**) shows the spatial and temporal scales of vector correlation during cluster movement. Scale bars, 10 μm.

concerted vertical oscillations in individual podosomes[23]. Therefore, our observation of directional flow for both proteins in podosome clusters raised the question as to whether there is cross correlation between these proteins. We imaged DCs co-expressing vinculin-GFP and LifeAct-RFP by confocal microscopy (Fig. 6a and Supplementary Movie 13) and the resulting time series were analysed with twSTICS that provided vector maps for both proteins individually (Fig. 6b). To quantify the degree of directional correlation between actin and vinculin vector maps within the podosome cluster, we calculated the directional correlation[30] of the actin and the vinculin vectors (Fig. 6c). A directional correlation value of one indicates perfect correlation, zero indicates perpendicular vectors, whereas minus

one indicates anti-correlated vectors. We found that the flows of vinculin and F-actin are largely correlated, as the mean directional correlation is almost equal to 1 (Fig. 6c). This is confirmed by comparison with randomized vectors, which we obtained by rearranging all the experimentally measured LifeAct-RFP vectors to a randomly selected spatiotemporal position within the podosome cluster and calculating the directional correlation between these randomized vectors and the original vinculin-GFP vectors. The mean directional correlation of these randomized vectors is almost 0, indicating complete lack of correlation, and is significantly different from the value obtained for the experimental vectors (Fig. 6c).

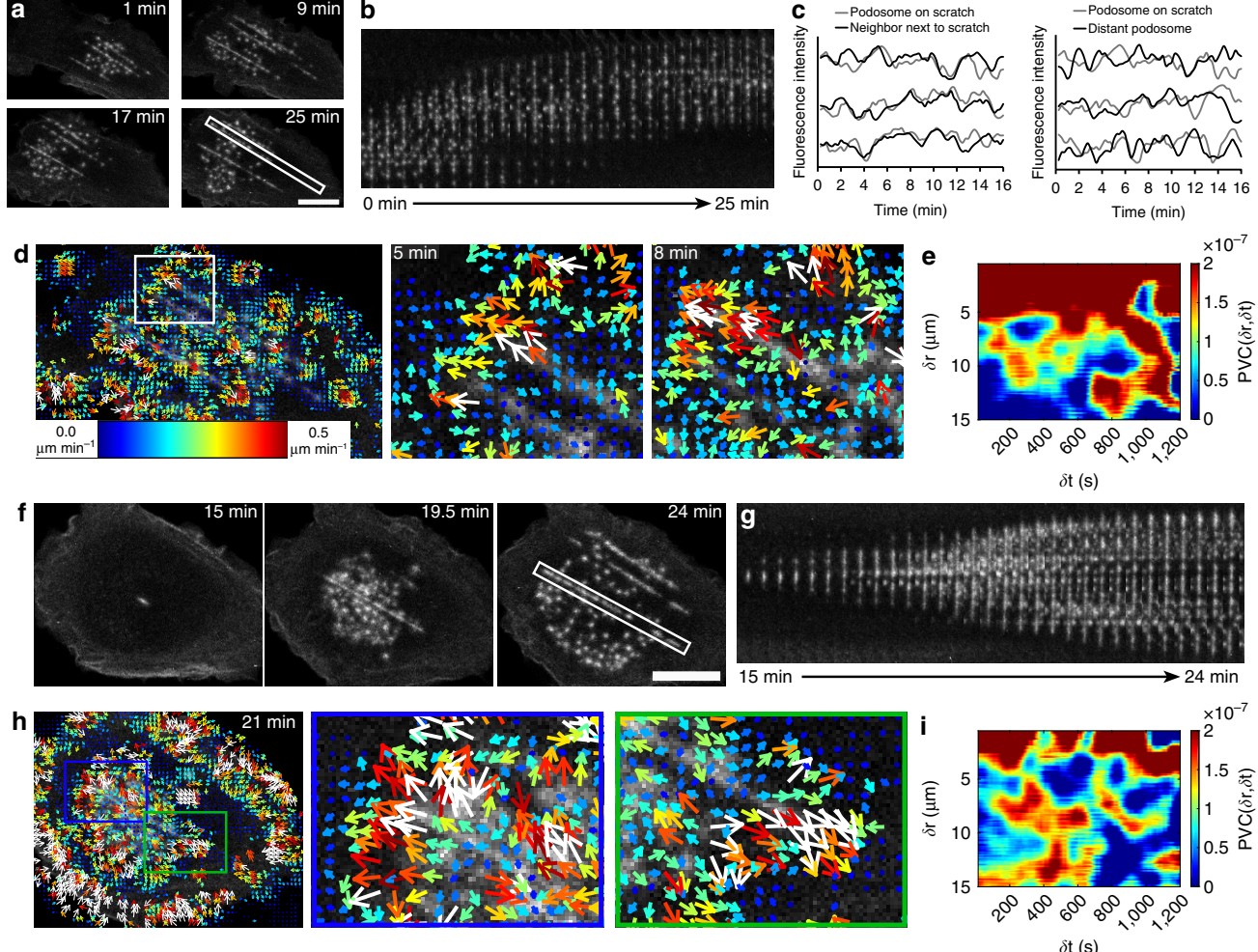

**Figure 5 | Podosome clusters respond to topographical cues.** (**a**–**e**) DCs were transfected with LifeAct-GFP and seeded on a patterned surface. Imaging was performed at a confocal microscope with 15 s frame intervals. Snapshots of the podosome cluster (**a**) and kymograph from white outlined area (**b**) show podosome cluster lateral movement. Plots of the GFP intensity fluctuations over time of three representative neighbouring and distant podosome pairs per condition (grey and black lines) are shown in **c**. Data are normalized to the mean intensity first, followed by normalization to minimum and maximum intensity. The time series was subjected to twSTICS analysis. Image and corresponding vector map plotted onto the immobile filtered version of the image are shown in **d**. The arrows in the vector maps indicate direction of flow and both the size and colour coding are representative of the flow magnitude. PVC plot (**e**) shows the spatial and temporal scales of vector correlation during cluster movement. (**f**–**i**) DCs were transfected with LifeAct-GFP and seeded on a patterned surface. Imaging was performed at a confocal microscope with 15 s frame intervals. Podosomes dissolution was induced by addition of 2–10 µM nocodazole, which was washed away when the dissolution was complete (not shown). The snapshots (**f**) and kymograph (**g**) show the part of the movie after nocodazole washout, when the podosomes reappear. Snapshots of the podosome cluster (**f**) and kymograph from white outlined area (**g**) show podosome cluster reformation, time stamps indicate time from the moment of washout. The time series (288 frames) was split into 3 overlapping series of 100 frames, which were subjected to twSTICS analysis. Image and corresponding vector map plotted onto the immobile filtered version of the image are shown in **h**. The arrows indicate direction of flow and both the size and colour coding are representative of the flow magnitude. The PVC plot (**i**) shows the spatial and temporal scales of vector correlation during cluster reformation on nocodazole washout. Scale bars, 10 µm.

Our previous studies showed almost no correlation between vertical oscillations of actin and talin in individual podosomes[23], and we therefore wondered whether talin also exhibited traveling waves throughout the podosome cluster and whether these were linked to the actin waves. First, we found that, similar to actin and vinculin, talin also exhibited dynamic spatial patterns within the podosome clusters as indicated by the twSTICS analysis of movies of DCs expressing talin-GFP (Supplementary Movie 14; Supplementary Fig. 5a). In DCs co-expressing talin-GFP and LifeAct-RFP, the degree of directional correlation between the talin and the actin vector is about 0.6, thus lower than what observed for vinculin and actin, but still significantly higher than what calculated for randomized vectors (Fig. 6d–f; Supplementary Movie 15). Interestingly, correlated flow between talin and vinculin was also observed as indicated by the cross-correlation twSTICS (twSTICCS, see Supplementary Methods) and the strong directional correlation among the vectors (Supplementary Movie 16; Supplementary Fig. 5b–e).

Therefore, we conclude that the traveling waves of vinculin, talin and actin exhibit spatiotemporal correlation with each other within podosome clusters, with vinculin patterns strongly correlated with both actin and talin patterns, and talin patterns modestly correlated with the actin patterns, as expected. These results emphasize the high degree of structural connectivity among podosomes belonging to the same cluster and put forward the podosome cluster as well-defined self-assembled signalling platform.

**Radiating actin filaments drive flow of podosome components**. Several types of lateral displacements are likely to occur for the actin structures within the podosome clusters, which can be identified by the twSTICS analysis: splitting and merging of podosome cores, podosome formation and dissolution, rearrangement and cross-linking of radiating filaments and actin polymerization and depolymerization both at the core and at the radiating filaments. To define the contribution of the core movements to the observed spatial patterns, we investigated the dynamics of WASP by confocal microscopy and subsequent twSTICS analysis (Supplementary Fig. 6 and Supplementary Movie 17). Since WASP specifically localizes at the base of the podosome actin core driving its assembly at the ventral membrane[34], we reasoned that WASP dynamics should primarily reflect the contribution to the twSTICS vectors coming from the short-distance displacements typical of the podosome cores[27]. As quantified in Supplementary Fig. 6b–e, WASP also exhibits clear albeit modest spatial patterns, as indicated by the low average velocity and limited spatiotemporal pair vector correlation, suggesting that the core movements contribute only partially to the overall spatial patterns of the podosome actin revealed by twSTICS analysis.

To determine the contribution of actin polymerization to the F-actin flow in the podosome cluster, we acquired time series of LifeAct-RFP expressing DCs on a confocal microscope before and after the addition of cytochalasinD (CytoD), which is a specific inhibitor of actin polymerization and disrupts the radiating actin filaments leaving the actin cores intact for the duration of our experiments (Fig. 7a). The resulting time series were analysed by twSTICS analysis (Supplementary Movie 18), and representative images of LifeAct-RFP and the corresponding vector maps before and after addition of CytoD are shown in Fig. 7b. Although some residual flow of F-actin was still present in the podosome cluster, there are many regions within the cluster where transport was hardly detectable following CytoD treatment. Moreover, the remaining vectors were less correlated with their neighbours in terms of direction and magnitude, suggesting that these vectors

represent noise rather than true molecular flow. To quantify this observation, we calculated the average flow magnitude in podosome clusters of cells before and after CytoD treatment and found a significant decrease in F-actin flow magnitude on inhibition of actin polymerization (Fig. 7c). Similar observations were made when Latrunculin A (LatA), another inhibitor of actin polymerization, was used (Supplementary Fig. 7 and Supplementary Movie 19), further confirming that actin polymerization plays a crucial role in generating the F-actin flow throughout the podosome cluster.

We next investigated whether inhibition of actin polymerization would affect the flow of other adaptor proteins present in the podosome. Since vinculin leaves the podosome cluster immediately after disruption of actin polymerization[9,23], we acquired time series of talin-GFP expressing DCs on a confocal microscope before and after the addition of CytoD and analysed them by twSTICS (Supplementary Movie 18). Images of talin-GFP with their corresponding vector maps before and after addition of CytoD are shown in Fig. 7d. Similar to F-actin, talin flow was largely reduced after CytoD treatment, as indicated by a lower average velocity (Fig. 7e). In addition, we investigated the fraction of ROIs that had a twSTICS vector (Fig. 7f) and generated histogram plots of the distribution of velocity magnitude values for actin before and after CytoD treatment (Fig. 7g). This showed that actin vectors had an overall homogeneous decrease in the fraction of vectors that passed the filtering criteria (Fig. 7f) but no decrease in the velocity magnitude distribution for the remaining vectors (Fig. 7g). For talin, we observed a slight increase in vector-free ROIs (Fig. 7h) and a clear shift of the velocity magnitude distribution towards lower velocities (Fig. 7i), meaning that after CytoD there are more talin vectors with slower speeds.

These results collectively demonstrate that dynamic patterns of both F-actin and adaptor proteins are strongly influenced by actin polymerization and require an intact actin filamentous network to exist and persist.

**Podosome clusters are self-assembling contractile platforms**. We have recently shown that although podosome vertical growth and shrinkage are dependent on myosin IIA mediated tension, the diffusion and binding kinetics of podosome components, as determined by FRAP, are not controlled by myosin II activity[23]. Although blocking of myosin IIA contractility has been shown to influence the protruding behaviour of individual podosomes by acting on the radiating filaments[23], a direct visualization of myosin II within the actin network of a podosome cluster was lacking so far. We therefore sought to investigate the reciprocal localization of actin and myosin IIA within podosome clusters at the nanoscale in DCs that were treated or not with myosin IIA inhibitor blebbistatin, fixed and stained for both proteins (Fig. 8a,b). By dual-color SIM, we here reveal that myosin IIA molecules specifically decorate almost all actin filaments radiating from the cores (Fig. 8a). Myosin IIA localization is retained also when the clusters form on patterned surfaces and nicely adapt to the overall structural rearrangement of the cluster caused by the topographical cue (Supplementary Fig. 8a,b). Treatment with blebbistatin clearly reduced the presence of myosin IIA and disturbed actin network integrity (Fig. 8b). Similarly, DCs treated with other inhibitors of myosin IIA activity such as ML7 (inhibitor of Myosin-Light Chain kinase) and Y27632 (ROCK inhibitor), fixed and stained for actin displayed altered actomyosin structures by SIM (Supplementary Fig. 8c). Despite their claimed specificity, these drugs could affect kinases other than MLCK or ROCK, which could also play a yet unknown role in podosome organization and dynamics.

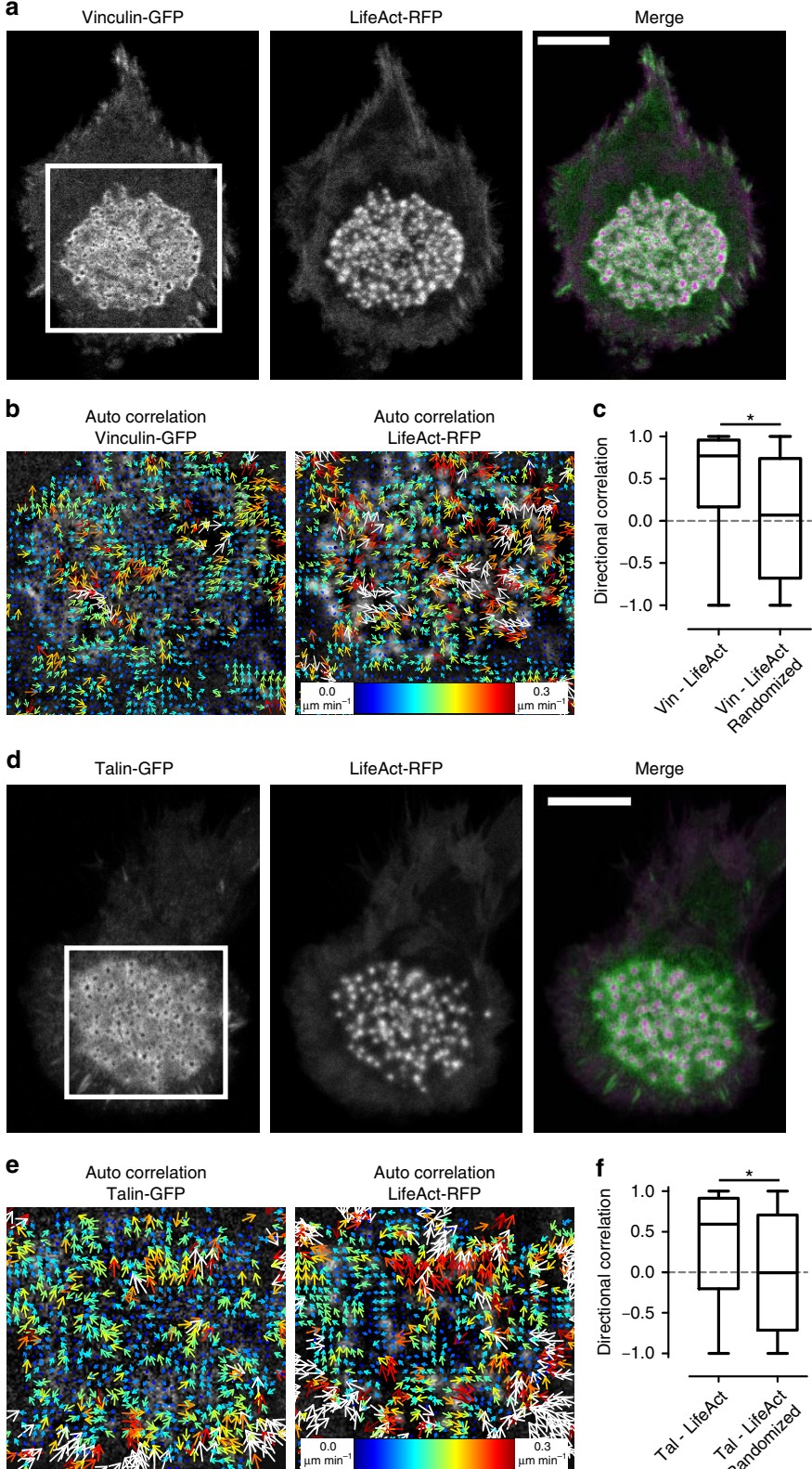

**Figure 6 | Flux of podosome core and ring components is directionally correlated.** DCs were transfected with LifeAct-RFP and cotransfected with vinculin-GFP (**a–c**) or talin-GFP (**d–f**) and seeded in a glass bottom dish. Imaging was performed at a confocal microscope with 15 s frame intervals. Shown are 10 frame moving average images of both channels (**a,d**). Time series for both channels (100 frames) were subjected to twSTICS, results are shown as vector maps in which the arrows indicate direction of flow and both the size and colour coding are representative of the flow magnitude. Vector maps are plotted onto the immobile filtered version of the image. Shown are the auto-correlation vector maps from the white boxed regions for both channels (**b,e**). Directional correlation was calculated between vectors measured by twSTICS in two different channels as the cosine of the angle between vectors with the same spatial and temporal lag (**c,f**). Asterisks indicated statistically significant differences (Mann–Whitney U-test, two-tailed, P<0.001 for **c** and **f**). Scale bars, 10 μm.

Since in our hands, overexpression of fluorescently tagged myosin IIA is not tolerated by DCs, we could not directly record myosin IIA dynamics in living cells. Therefore, to determine whether myosin IIA activity would influence the mesoscale spatial patterns observed for the podosome components, we recorded the dynamics of LifeAct-RFP in DCs treated with blebbistatin (Supplementary Movie 20). TwSTICS analysis of these movies showed an overall decrease in the velocity magnitude values, as visually indicated in the snapshots of the twSTICS analysis (Fig. 8c) and quantified in the average velocity plot (Fig. 8d).

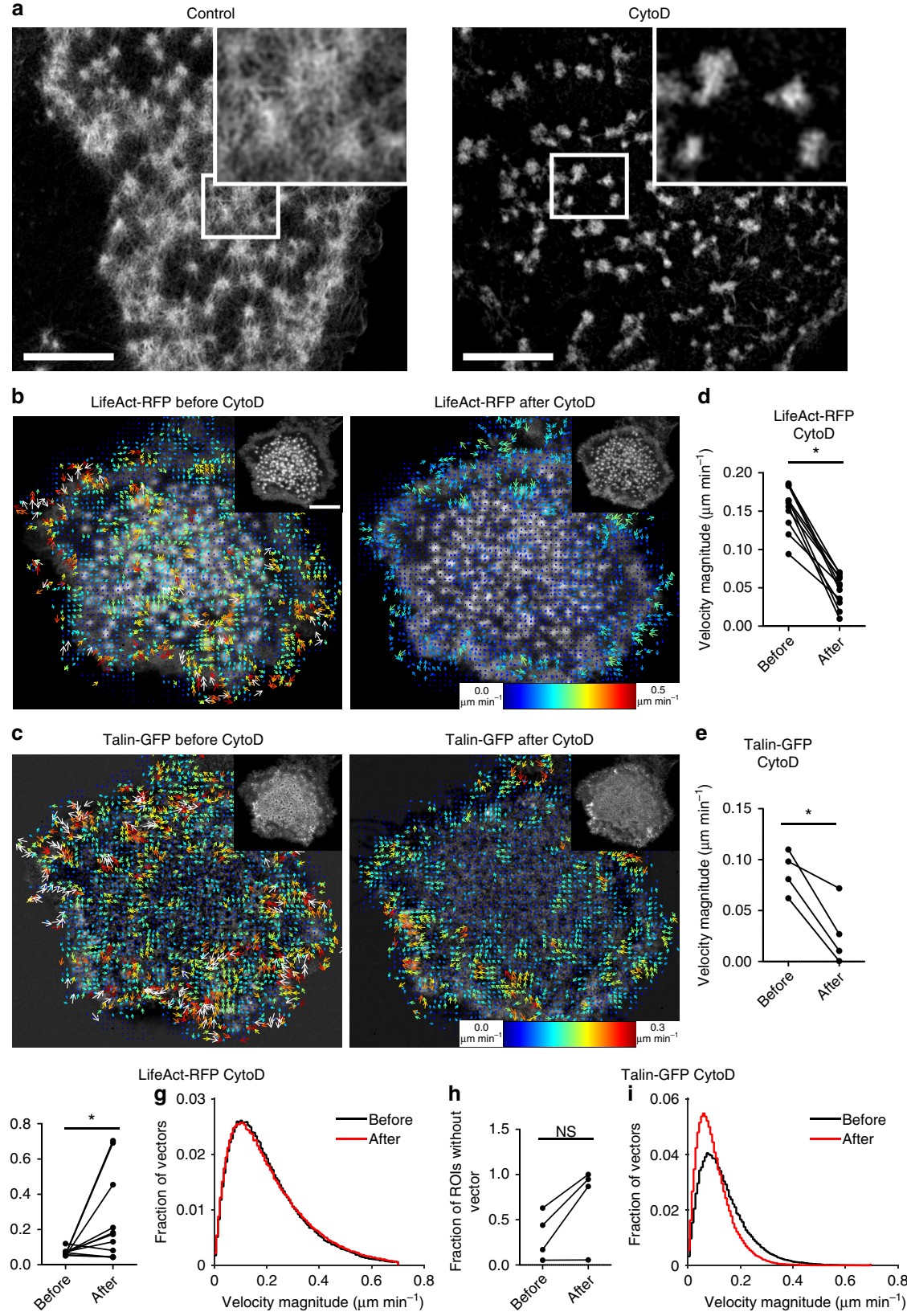

Furthermore, Fig. 8e shows a slight increase in ROIs without detectable vectors post blebbistatin treatment, and the histogram of the distribution of velocity magnitude values for control cells versus cells treated with blebbistatin also displays a shift of the distribution of magnitude values towards lower velocities (Fig. 8f). This means that after blebbistatin treatment, fewer vectors and slower speeds are found for actin.

These data collectively demonstrate the importance of an intact and functional actomyosin network at the podosome cluster to preserve and drive the dynamic spatial patterns of the different podosome components and confirm the concept that a podosome cluster is a contractile platform well suited for mediating DC adhesion and environment sensing.

## Discussion

By integrating nanoscopy of multiple podosome components with measurements of their spatiotemporal molecular dynamics using an extension of STICS and with the application of dedicated image analysis algorithms, we here revealed the dynamic and structural mechanisms driving mesoscale coordination of podosome clusters in human DCs. We demonstrated that F-actin, vinculin and talin exhibit previously unappreciated directed flow patterns in podosome clusters (Fig. 9 and Supplementary Movie 21). These dynamic spatial patterns correlate with the presence of intact actin filaments enriched in myosin IIA and interconnecting multiple adjacent podosomes within the cluster. Both spatial patterns and structural connections depend on actin polymerization and myosin IIA contractility and provide a continuous feedback loop for these adhesive and protrusive structures to respond to changes in substrate topography thus mediating DC mechanosensing.

We have previously shown that talin, vinculin, zyxin and paxillin are highly dynamic at the level of individual podosomes[23]. Here we demonstrated the existence of dynamic spatial patterns involving multiple adjacent podosomes and developed an approach to quantify the directionality of these cytoskeletal components within the podosome cluster. While the rapid diffusion of podosome structural components at the level of individual podosomes was not dependent on myosin IIA contractility[23], the mesoscale directionality is, as blocking myosin IIA significantly decreases the spatiotemporal extent of this phenomenon. This most probably relates to a global loss of podosome cluster dynamics, rather than a change in diffusion and binding kinetics of individual components. Myosin IIA dependent contraction is therefore essential to control the coordinated redistribution of cytoskeletal components throughout the cluster.

Within the podosome cluster, actin exhibits various dynamic behaviours, ranging from polymerization at the base of the branched actin core, polymerization at the radiating actin filaments as well as mesoscale directionality, posing the fascinating question of how the actin dynamics at the single podosome level relate to the cluster-broad actin dynamic patterns. Destaing and colleagues previously performed FRAP analysis of a small area covering multiple podosomes within one cluster and reported that the recovery of the actin cloud and core actin were highly correlated[18]. The authors proposed that the two actin structures must be dynamically correlated, and that the actin cloud derives from the actin core[18]. Our observation that photoconverted-GFP-tagged vinculin showed a preferred directionality within the podosome cluster supports the notion that a direct mechanical connection may exist between the various podosome substructures, as actin and actin-interacting proteins are redistributed throughout the cluster.

Coordinated dynamics within podosome clusters have been described in macrophages, where old podosomes dissolve at the cell interior and new podosomes are formed at the leading edge[24,35]. Moreover, in osteoclasts, podosomes are initially formed as small clusters, which then collectively coalesce to form a ring-like structure that eventually becomes a belt[18]. Our study supports and further extends these observations providing novel mechanistic insights that explain how individual podosomes communicate and transmit mechanical and biochemical information within the same cluster. By combining time-lapse PFM with finite element simulations, Poincloux and colleagues recently demonstrated the existence of spatial synchrony of force dynamics as well as actin content between two neighbouring podosomes[27]. This group and we have previously established that increased actin content at the core corresponds to increased protrusion forces towards the underlying substrate[23,36]. These reports agree with our current findings that neighbouring podosomes exhibit correlated fluctuations of actin content, which are not detected in distant podosome pairs. However, twSTICS and PVC analysis allowed us to expand these observations and discover that directional spatiotemporal coordination of cytoskeletal components exists between podosomes at large spatial scales (6–8 μm), which extends beyond nearest neighbours spanning multiple proximal podosomes, and at temporal scales up to 15 min. This long-range spatiotemporal correlation is clearly much more pronounced in conditions such as de novo cluster formation or collective displacement of a podosome cluster as observed during cell movement. It remains to be determined how individual podosome oscillations are exactly linked to the waves we describe here. We hypothesize that spatial coordination of actin and force oscillations results in a wave like phenomenon, which could lead to oscillatory recruitment of mechanosensitive proteins like vinculin to adjacent podosome. The fact that inhibition of either actin polymerization or myosin IIA diminishes force and F-actin oscillations[23,36] as well as flow further confirms the link between vertical oscillations and dynamic spatial patterns. PFM measurements of individual podosomes on substrates of different stiffness demonstrated that each podosome generates a

**Figure 7 | Actin polymerization and network integrity are essential for podosome core and ring flux.** (**a**) DCs adhering to a glass coverslip were left untreated or treated with 2.5 μg ml⁻¹ CytoD for 10 min, fixed, permeabilized and labelled for actin by phalloidin. Samples were mounted in mowiol and imaged by SIM. (**b-i**) DCs were co-transfected with LifeAct-RFP and talin-GFP and seeded in a glass bottom dish. Imaging was performed at a confocal microscope with 15 s frame intervals, after 50 frames 2.5 μg ml⁻¹ CytoD was added and imaging was continued up to 100 frames. Time series were subjected to twSTICS analysis, auto-correlation results are shown for LifeAct (**b**) and talin (**c**). Images and corresponding vector maps plotted onto the immobile filtered version of the image are shown before and after addition of CytoD. The arrows indicate direction of flow and both the size and colour coding are representative of the flow magnitude. Quantification of flow magnitudes before and after addition of CytoD is shown for LifeAct (**d**) and talin (**e**), where each dot represents a single cell and lines connect the same cells before and after treatment. Fraction of ROIs within the podosome cluster without a vector before and after addition of CytoD is shown for LifeAct (**f**) and talin (**h**), where each dot represents a single cell and lines connect the same cells before and after treatment. Pooling all the cells used for panels **d,e,f** and **h**, histograms of flow velocities are shown for LifeAct (**g**) and talin (**i**) before and after the addition of CytoD. Asterisks indicated statistically significant differences (paired student t-test, two tailed, P = 4.12e−7 (**d**), P = 0.0157 (**e**), P = 0.0406 (**f**), P = 0.0752 (**h**)). Scale bars represent 10 μm. ns, not significant.

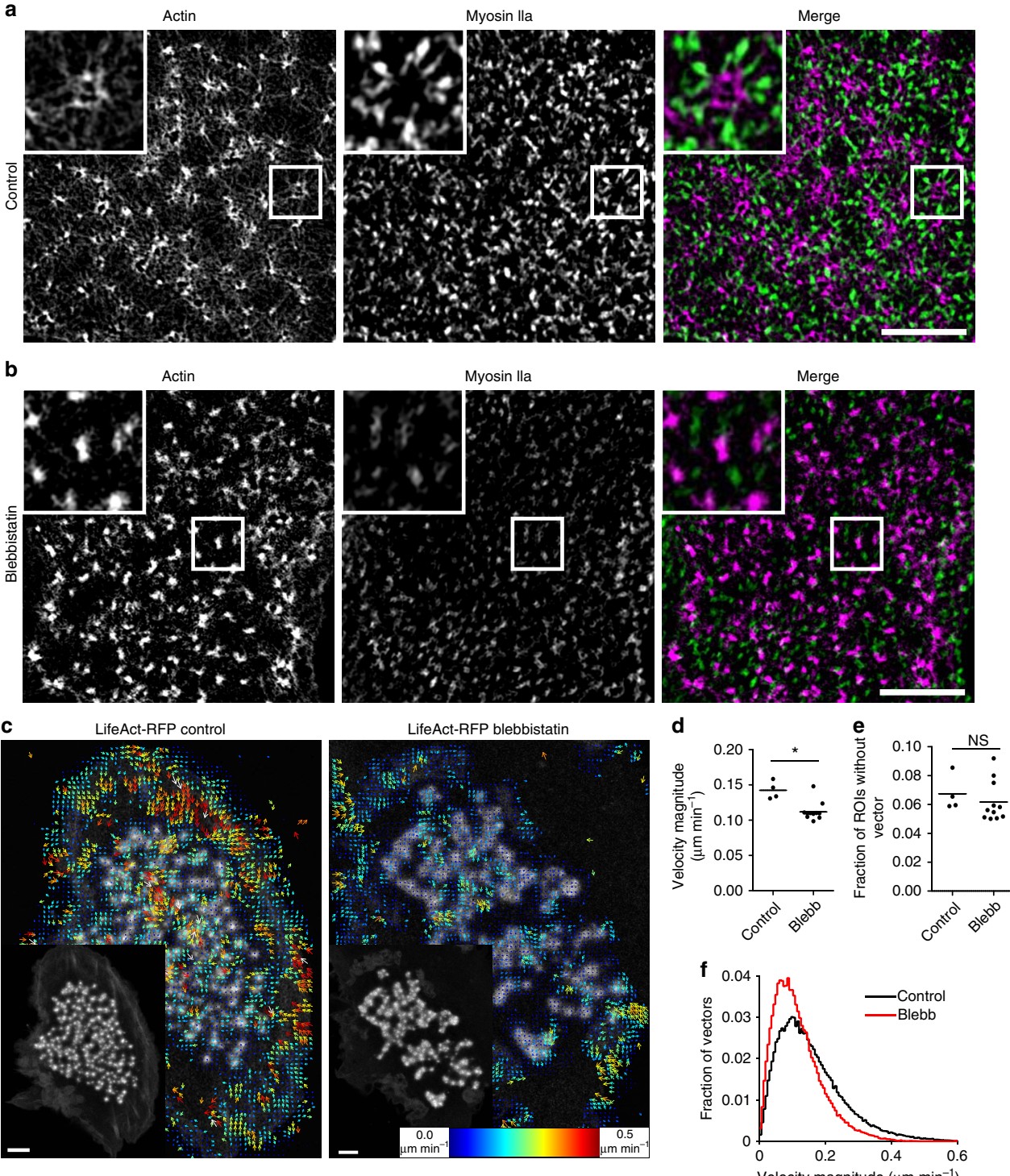

**Figure 8 | Myosin IIA-mediated contractility is essential for actin flux.** DCs adhering adherent to a glass coverslip were left untreated (**a**) or treated with 50 μM blebbistatin for 60 min (**b**), fixed, permeabilized and labelled for actin (magenta) and myosin IIa (green) by fluorescent phalloidin and anti-myosin IIA mAb, respectively. Samples were mounted in mowiol and imaged by SIM. One representative cell is shown. (**c**–**f**) DCs were transfected with LifeAct-RFP and seeded in a glass bottom dish. Cells were left untreated or pretreated for 60 min with 20 μM blebbistatin. Imaging was performed at a confocal microscope with 15 s frame intervals. Time series (100 frames) were subjected to twSTICS analysis, results are shown as vector maps in which the arrows indicate direction of flow and both the size and colour coding are representative of the flow magnitude (**c**). Vector maps are plotted onto the immobile filtered version of the images. A 10 frame moving average corresponding to the STICS window is shown in the lower left insets. Quantification of flow magnitudes for untreated cells and cells treated with blebbistatin is shown (**d**), where each dot represents a single cell and the line shows mean. Fraction of ROIs without a vector within the podosome cluster is plotted in (**e**) where each dot represents a single cell and the line shows mean. Pooling all the cells used for **d**,**e**, histograms of velocity magnitude values are shown in (**f**). Asterisk indicates statistically significant difference (paired student t-test, two-tailed (**d**) P = 0.0013, (**e**) P = 4947). One representative movie from three separate experiments is shown. Scale bars represent 10 μm. ns, not significant.

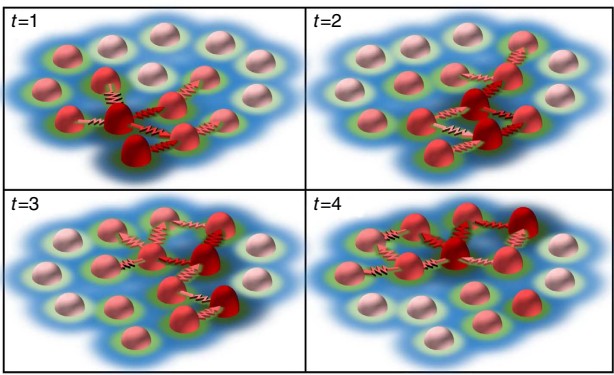

**Figure 9 | Schematic representation of the dynamic spatial patterns within podosome clusters.** Individual podosomes (light to dark red cones) continuously undergo growth and shrinkage cycles. Within a cluster, podosome neighbours are connected by myosin decorated actin filaments (red arrows) that drive a spatial coordination of actin and force oscillations resulting in a wave-like phenomenon ($t = 1$ to $t = 4$, arbitrary time units). These continuous actin waves directly drive the recruitment of vinculin (green) and talin (blue) resulting in mesoscale dynamic spatial patterns within the cluster that extend well beyond the nearest neighbour. For simplicity, the actin filaments connecting individual podosomes to the cell membrane are not depicted.

protrusion force that increases with the stiffness of the substratum, which is a typical behaviour of mechanosensing structures[36]. The mesoscale spatiotemporal correlation revealed by our study puts forwards the intriguing concept that propagation of mechanosensing could occur over multiple podosomes and for prolonged time, demonstrating that the podosome cluster can actually act as a dynamic mechanosensing platform in tissue-resident DCs.

Dynamic spatial patterns of actin and actin-interacting proteins are emerging as important regulatory mechanisms of a large variety of cellular processes including cell division[37], polarity[38], activation[39] and motility[40]. Traveling waves of signalling molecules or macromolecular complexes are thought to propagate cellular signals and organize cell responses. At the cell ventral membrane, waves of actin have been identified that represent novel integrin-based adhesion complexes in fibroblasts and melanoma cells[7] as well as F-actin waves formed by neutrophils during chemotactic migration[41]. Of special interest is the recent observation that Cdc42 plays a key role in the generation of F-actin waves in mast cells where a transition from traveling to standing waves of actin polymerization has been observed[39]. In the context of podosome dynamics, the podosome protrusion fluctuations resemble the actin standing waves, while the flows of components observed here by twSTICS are reminiscent of the traveling waves. Cdc42 plays a crucial role in podosome formation by interacting with WASP at the podosome core[34]. Although direct evidence is still lacking, Cdc42 most likely exhibits a dynamic pattern profile similar to WASP. Here we show that myosin IIA contractility is essential for the F-actin flow, pointing towards a possible role for RhoA, key regulator of myosin IIA activity, in modulating the flows of podosome components. A local balance among RhoA, Cdc42 and Rac activities has been shown to regulate podosome formation, dissolution and dynamics[42–45] and might therefore function upstream of the observed waves for F-actin, vinculin and perhaps also talin. Capturing signalling waves of these small GTPases within the podosome cluster is a challenge that future investigation should address to fully unravel the molecular mechanisms behind the mesoscale dynamic spatial patterns.

TwSTICS applied to imaging of small GTPase activity by next generation FRET-based biosensors could provide a useful tool for this purpose.

Quantifying irregular dynamic patterns of proteins inside the cell is challenging. Previously, STICS has been applied to systems with relatively slow dynamics by averaging over many image frames[30,46]. This long time averaging does not permit resolution of transient flow patterns in more 'unstable' dynamic systems like podosomes and podosome clusters. Here we calculated the vector maps from short sequences of 10 image frames, corresponding to a 2.5 min period. At this time scale, podosome clusters are relatively stable entities (i.e., do not significantly assemble or disassemble), although intensity fluctuations at the level of individual podosomes still occur[23]. By applying twSTICS, we were therefore able to study the time evolution of vector maps and to capture the rapid dynamics of several components throughout a podosome cluster.

TwSTICS could potentially also be applied to study even faster dynamics when combined with wide field or TIRF microscopy which allow frame rates up to 20 fps. However, at these imaging rates the on-off kinetics of probes like LifeAct-GFP become relevant because they occur at the same time scale as the process studied. In that case the use of actin-GFP or other F-actin binding probes with different on-off kinetics should be considered. Also twSTICS spatial resolution can be improved by performing STICS analysis of live cell super-resolution data[47]. Both twSTICS and PVC analysis of our data have provided quantification of velocities and spatiotemporal correlation scales within podosome clusters. These variables, combined with information about podosome nanoarchitecture[9] and other parameters such as diffusion and binding kinetics of components[18,23] and protrusion forces[25,27,36] could be used to construct or expand theoretical models of podosome dynamics and forces[48]. It would be of great interest to test model and simulation predictions using our methods, which would deepen our mechanistic understanding of the regulation of mesoscale dynamics. Notably, PVC combined with twSTICS is a powerful tool to reveal the additional information on the relevant correlation length and time scales not previously accessible by the original STICS.

New approaches that combine functional imaging with techniques that report on molecular mobility have been acknowledged as a prerequisite for obtaining a deeper understanding of the mechanisms that drive spatiotemporal patterns in living cells[49]. Our combination of twSTICS with PVC is a significant step in this direction as we demonstrated in the quantification of podosome cluster mesoscale dynamics. Ongoing efforts look to extend this to map signalling dynamics in reference to the cellular structures.

Our findings put forward the concept that the podosome cluster is a highly dynamic mechanosensory platform, which provides DCs with a substrate-probing surface where mechanical feedback rapidly redistributes and possibly regulates local secretion of matrix degrading proteases. A relationship between podosome formation and mechanical properties of the cell environment very recently emerged for endothelial cells[50] and muscle cells[51], which strongly motivate further investigation on podosome-mediated mechanosensing. Dynamic spatial patterns of cytoskeletal components may also exist in other cell types that exhibit mesoscale organization of podosomes, such as osteoclasts[18,26] and endothelial cells[52]. Podosomes are strongly related to other actomyosin-based structures such as invadopodia[53]. Invadopodia mediate physiological breaching of the basement membrane during *C. elegans* larval development[54]. Furthermore, invadopodia facilitate migration and proteolytic tissue invasion of cancer cells[12,55] and have been recently proposed as novel therapeutic target to prevent cancer

metastasis[56]. A better understanding of the architecture and dynamic behaviour of these actin-based structures will advance our fundamental understanding of cell mechanosensing strategies while providing novel leads to improve specificity and efficacy of experimental anti-cancer therapies. Our study therefore provides a predictive model for mesoscale structural and dynamic organization of cytoskeletal structures testable in a variety of cell types.

## Methods

**Preparation of human DCs.** DCs were generated from peripheral blood mononuclear cells[57,58]. Monocytes were derived either from buffy coats or from a leukapheresis product. Peripheral blood mononuclear cells (PBMCs) were isolated by Ficoll density gradient centrifugation (GE Healthcare Biosciences, 30 min, 4 °C, 2,100 r.p.m.). PBMCs were extensively washed in cold phosphate buffered saline (PBS) supplemented with 0.1% (w/v) bovine serum albumin (BSA, Roche Diagnostics) and 0.45% (w/v) sodium citrate (Sigma Aldrich). PBMCs were seeded in plastic culture flasks for 1 h and monocytes were isolated by plastic adherence. Monocytes were cultured in RPMI 1640 medium (Life Technologies) supplemented with fetal bovine serum (FBS, Greiner Bio-one), 1 mM Ultra-glutamine (BioWhittaker), antibiotics (100 U ml$^{-1}$ penicillin, 100 μg ml$^{-1}$ streptomycin and 0.25 μg ml$^{-1}$ amphotericin B, Gibco) for 6 days, in a humidified, 5% $CO_2$-containing atmosphere. During these six days DC differentiation was induced by addition of IL-4 (500 U ml$^{-1}$) and GM-CSF (800 U ml$^{-1}$) to the culture medium. At day 5 or day 6 cells were collected and reseeded onto coverslips or imaging dishes.

**Antibodies and reagents.** The following primary antibodies were used: anti-myosin IIA (#PRB-440P, Biolegend, 1:200 dilution), anti-vinculin (#V9131, Sigma-Aldrich, 1:400 dilution). Secondary antibodies conjugated to Alexa647 were used (Life Technologies, 1:400 dilution). F-actin was stained with Alexa488-conjugated phalloidin (#A12379, Life Technologies, 1:200 dilution).

The following inhibitors were used: cytochalasin D (2.5 μg ml$^{-1}$, Sigma-Aldrich), blebbistatin (20–50 μM, Sigma-Aldrich), nocodazole (2–10 μM, Sigma-Aldrich), ML7 (10 μM), Y27632 (20 μM), and LatrunculinA (0.5 μM, Sigma-Aldrich).

**Immunofluorescence.** Cells were seeded on glass coverslips (EMS) and left to adhere for 4 h. To image cells on scratched glass, coverslips were rubbed (3 strokes) over abrasive paper of P180 grit size before sterilization and cell seeding. Cells were fixed in 4% (w/v) paraformaldehyde in PBS for 15 min after which they were permeabilized in 0.1% (v/v) Triton X-100 in PBS for 5 min and blocked with 3% (w/v) BSA, 20 mM Glycine and 1% normal donkey serum in PBS. The cells were incubated with primary Ab for 1 h. Subsequently, the cells were washed with PBS and incubated with secondary antibodies and phalloidin for 45 min and washed with phosphate buffer before embedding in Mowiol (Sigma-Aldrich).

**Structured illumination microscopy.** Structured illumination imaging was performed using a Zeiss Elyra PS1 system. 3D-SIM data was acquired using a 63 × 1.4 NA oil objective. 488, 561, 642 nm 10 mW diode lasers were used to excite the fluorophores together with, respectively, a BP 495–575 + LP 750, BP 570–650 + LP 750 or LP 655 excitation filter. For 3D-SIM imaging the recommended grating was present in the light path. The grating was modulated in 5 phases and 5 rotations, and multiple z-slices with an interval of 110 nm were recorded on an Andor iXon DU 885, 1,002 × 1,004 EMCCD camera. Raw images were reconstructed using the Zeiss Zen 2012 software. For SIM representative images, the following number of experiments were performed: (i) glass versus scratches, three experiments, per experiment five images from different cells for each condition; (ii) CytoD versus untreated, three experiments, per experiment five images from different cells for each condition; (iii) myosin IIA inhibition versus untreated: two experiments (one with Blebb and one with ML7 + Y27632 mix), five images from different cells for each condition.

**DC transfection.** Transient transfections were carried out with the Neon Transfection System (Life Technologies). Cells were washed with PBS and resuspended in 115 μl Resuspension Buffer per 0.5 × 10$^6$ cells. Subsequently, cells were mixed with 6 μg DNA per 10$^6$ cells per transfection and electroporated. Directly after, cells were transferred to WillCo-dishes (WillCo Wells B.V.) with pre-warmed medium without antibiotics or serum. After 3 h, the medium was replaced by a medium supplemented with 10% (v/v) FCS and antibiotics. Before live-cell imaging, cells were washed with PBS and imaging was performed in RPMI without Phenol red. All live cell imaging was performed at 37 °C.

**Live cell imaging.** Transiently transfected cells were imaged on various confocal microscopes, but acquisition parameters were constant for all images: 140 nm pixel diameter with frames acquired every 15 s at 37 °C, with the only exception being the movies for Fig. 1a,b, see below for details. For dual color imaging experiments,

images were acquired sequentially to prevent signal bleed-through. For live-cell imaging experiments with blebbistatin, cells were transfected with lifeact-RFP and exclusively excited with red (>540 nm) light to prevent photoinactivation[59] and phototoxicity[60] of blebbistatin by blue excitation light. The following confocal microscopes were used:

- Zeiss LSM 510, equipped with a PlanApochromatic 63 × /1.4 NA oil immersion objective. The samples were excited with a 488-nm argon (GFP) and/or a 543-nm HeNe (RFP) laser lines.
- Leica SP5 (1), equipped with a HCX PL APO CS 63 × /1.2 NA water immersion objective and an acousto-optical beam splitter (AOBS). mCherry was excited with a 594 nm laser line and emission light was filtered by the AOBS set at 600–700 nm.
- Leica SP5 (2), equipped with an HCX PL APO CS 63 × /1.4 NA oil objective and an (AOBS). GFP was excited by 488 nm argon laser line and emission light was filtered by the AOBS set at 500–600 nm. RFP was excited by 561 nm laser line and emission light was filtered by the AOBS set at 570–700 nm.
- Leica SP8, equipped with a HC PL APO CS2 63 × /1.2 NA water immersion objective, a supercontinuum white light laser (WLL), a 488 nm argon laser, an AOBS and Hybrid detectors (HyD). All images were collected using HyD detectors operated in photon counting mode with time gating (0.3–6 ns) to minimize background from laser reflections. For dual color GFP-RFP experiments GFP was excited by 488 nm light and emission light was filtered by the AOBS set at 500–550 nm, RFP was excited by 540 nm light and emission light was filtered by the AOBS set at 570–700 nm, images were acquired sequentially to prevent bleed through. For dual color GFP-mCherry images GFP was excited by 488 nm light and emission light was filtered by the AOBS set at 494–559 nm, mCherry was excited by 594 nm light and emission light was filtered by the AOBS set at 603–689 nm, images were acquired simultaneously.

Live cell imaging for Fig. 1a,b was performed on a LeicaDMI6000 epi-fluorescence microscope equipped with an HC PL APO 63 × /1.40–0.60 oil objective. GFP was excited with a metal halide lamp through a 470/40 nm band pass filter and emission was detected through a 524/50 nm band pass filter. Imaging was performed at 37 °C. Scratches were made as described in the immunofluorescence section. Both on glass and scratches >10 cells were imaged, and one representative movie is shown.

**Imaging tdEos-Vinculin.** DCs expressing tdEos-Vinculin were imaged on a Leica SP8 confocal microscope (see above). tdEos was excited with the WLL set at 506 nm (green) and 570 nm (red). Emission light was filtered by the AOBS set at 510–560 nm (green) and 575–700 nm (red). Data were acquired sequentially for the red and green channels. Photoswitching was performed in a 6 μm circular ROI by a 405 Diode laser operated at maximum power. Seven cells from 2 independent experiments were imaged and analysed of which one representative example is shown.

**Fluorescence recovery after photobleaching.** DCs expression either LifeAct-GFP or actin-GFP were imaged on a Leica SP8 confocal microscope (see above). FRAP experiments were performed using a 2.1-μm diameter circular region of interest in individual podosomes. Photobleaching was performed operating the argon laser at 100% of laser power by scanning the bleached ROI for two iterations, yielding a total bleach time of 0.10 s and an average fluorescence loss of ∼50%. Recoveries were collected with time intervals of 100 ms (LifeAct-GFP) or 300 ms (Actin-GFP) using 488 nm WLL excitation. 12 cells were imaged for both proteins and shown are the FRAP curves averaged for those 12 cells.

**Image analysis.** All image analysis was performed using Fiji[61]. Before analysis movies that showed signs of stage drift were registered allowing translation only, using the StackReg plugin[62] and bleach correction was performed. For fluctuation analysis podosome intensity was measured in a circular ROI with a 1.4 μm diameter. In the Supplementary Movies of confocal time series with corresponding vector maps, the confocal time series is shown as a moving average of 10 frames to allow direct comparison with the corresponding vector map. Movies without vector maps are shown as a moving average of three frames to suppress noise.

**Time window STICS analysis.** We performed STICS[46] with a short time window iterated in single frame shifts on a CLSM time series of fluorescently tagged LifeAct, vinculin and talin, acquired with a 15 s time lag between frames. First, a Fourier immobile filter was applied in time to each pixel stack in the entire image series to remove the lowest frequency (for example, static) components[46]. Subsequently, each image was divided into 16 × 16 pixels ROIs (2.24 × 2.24 μm) and adjacent ROIs were shifted four pixels in the horizontal and vertical directions to map the entire field of view with oversampling in space. The time series was divided into overlapping 10 frame sized TOIs (2.5 min) and adjacent TOIs were shifted one frame for each STICS analysis to cover the entire image series with oversampling in time. Space-time correlation functions were calculated for each ROI/TOI and fit for time lags up to τ = 8 to measure vectors (magnitude and

direction) of the flow from the translation of the correlation peak as described earlier[30] (see also Supplementary Methods). Detected noise vectors, which are due to random fits to spurious background peaks that pass multiple fitting threshold criteria, become more significant as we reduce the statistical sampling with short time windows. However, noise vectors exhibit little correlation with their neighbours in terms of direction and magnitude for systems where there are real flows. Due to the spatial and temporal oversampling (75% common overlap in space between adjacent ROIs and 90% common overlap in time between sequential TOIs) we expect neighbouring vectors to correlate in magnitude and direction for real flows. Noise vectors that pass the fitting criteria were eliminated by setting a vector similarity criterion for adjacent vectors. All retained vectors were plotted on the corresponding frames of the immobile filtered image series. Further details about the twSTICS method are described in the Supplementary Methods and Supplementary Fig. 2. For representative images shown of twSTICS experiments the following number of cells were imaged and analysed: LifeAct untreated: >30 cells, LifeAct before/after CytoD: 11 cells, LifeAct before/after LatA: 11 cells, LifeAct after Blebb treatment: 11 cells + 4 untreated controls cells imaged on same day, talin untreated: 29 cells, talin before/after CytoD: 4 cells, vinculin untreated: 26 cells, actin-GFP: 6 cells, WASP untreated: 9 cells, LifeAct on patterned surfaces: 5 cells, and 3 cells before and after nocodazole washout on plain glass and 3 cells before and after nocodazole washout on patterned surfaces. We recorded six movies of spontaneously moving clusters in scratches (LifeAct-GFP) and three movies of spontaneously moving clusters on flat glass (1 LifeAct-GFP and 2 Vinculin-GFP). These images were collected from at least two similar experiments for each condition.

**Vector velocity magnitude quantification.** Velocity magnitudes were quantified inside podosome clusters. First, the original confocal image sequence was filtered with a 10 frame moving average filter to match the twSTICS data 10 frame TOI. Next, the sequence was filtered with a Gaussian filter and an isodata intensity threshold was applied to select the podosome cluster region and serve as a mask to select vectors within the cluster only. To calculate the mean velocity magnitude per cell, ROIs without a vector were considered to have a zero velocity. The fraction of ROIs without vector were calculated per cell and histograms were calculated by combining data for all cells analysed.

**Vector directional correlation.** A correlation coefficient was calculated as the cosine of the angle between two vectors with the same spatial and temporal lags that had been measured by twSTICS in two different detection channels. This directional coefficient is a number that varies between $-1$ (vectors pointing in the opposite direction) and 1 (vectors pointing in the same direction), with perpendicular vectors yielding 0. To verify the significance of the calculated distribution of directional correlation coefficients we compared with a directional coefficient calculated from a random distribution of uncorrelated vectors. This was done by assigning a random position in space and time within the time series to each vector from one channel and then calculating the directional correlation coefficient between these random vectors with the original unmodified vectors from the other detection channel. In all cases this resulted in a correlation coefficient distribution with a median close to zero. For twSTICS with directional correlation the following number of cells were analysed: LifeAct-talin: four cells, LifeAct-vinculin: six cells, vinculin-talin: four cells. These movies were collected from at least two similar experiments for each condition.

**Pair vector correlation.** To determine the spatial and temporal scales over which flow is correlated within a podosome cluster, we calculated the dot product between vectors separated in space and time (Fig. 3c). We calculated an average PVC function for all pairs of vectors separated by the same spatiotemporal lags according to:

$$\text{PVC}(\delta r, \delta t) = \frac{1}{M_{\text{pairs}}(\delta r, \delta t)} \sum_i \sum_j \mathbf{v}_i(r, t) \cdot \mathbf{v}_j(r + \delta r, t + \delta t) \qquad (1)$$

where $\delta r$ and $\delta t$ are the radial spatial and temporal lags, $M_{\text{pairs}}(\delta r, \delta t)$ denotes the number of vector pairs for each specified spatio-temporal lag and $\mathbf{v}_i$ and $\mathbf{v}_j$ are the vector pairs multiplied as dot products. When the angle between the two vectors lies between $-90°$ and $90°$, the dot product is positive. Conversely, when the angle between the two vectors is between $-90°$ and $-180°$ or $90°$ and $180°$, the dot product is negative. When the vectors are uncorrelated the PVC will average to zero. For representative PVC experiments the following number of cells were analysed: LifeAct stationary clusters: 6 cells, vinculin stationary clusters: 4 cells, Moving clusters: 4 LifeAct cell, 2 vinculin cells, Cluster reformation: 10 LifeAct cell after nocodazole washout, 1 LifeAct cell with spontaneous cluster formation, WASP: 6 cells. These analyses were carried out on all movies analysed by twSTICS.

**Statistical analysis.** Statistical analysis was carried out with GraphPad Prism, Microsoft Excel and Matlab. Data are presented as median ± interquartile range with 10 and 90 percentile whiskers for box plots. A Student's $t$-test or Mann–Whitney $U$-test was used for comparison of two groups. Statistical significance was defined as $P < 0.05$.

**Data availability.** All relevant data is available from the authors. All computer code is available by contacting Paul W. Wiseman.

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

## Acknowledgements

We are greatly indebted to the following people who kindly provided us the plasmids used in this manuscript: Johan de Rooij (vinculin-GFP); Anna Huttenlocher (talin-GFP); Michael Sixt (Lifeact-GFP and -RFP); Jennifer Gillette (tdEos-vinculin); Stefan Linder (WASP-GFP); Victor Small (vinculin-mCherry); Kenneth Yamada (talin-mCherry). The authors also thank the Microscopic Imaging Center of the Radboud Institute for Molecular Life Sciences and the Erasmus Optical Imaging Center for use of their microscopy facilities. This work was financially supported by EU grant NANOVISTA (2882630) and by a Human Frontiers Science Program Grant (RGP0027/2012) to A.C. as well as a Natural Sciences and Engineering Research Council of Canada Discovery Grant to P.W.W. S.M. is supported by an intramural PhD fellowship from the Radboud University Medical Center. We are indebted to Stefan Linder, Peter Friedl and Jack Fransen for insightful discussion and critical reading of the manuscript.

## Author contributions

M.B.M.M., E.P., S.F.B.M., K.V.D.D., B.J., L.M.P. and J.A.S. performed experiments. M.B.M.M., E.P., J.A.S. and B.J. carried out data analysis. P.W.W., E.P., D.G., J.A.S. and A.B.H. provided analytical tools. M.B.M.M., K.V.D.D., P.W.W. and A.C. designed the study. A.C. supervised the entire study. M.B.M.M., K.D., P.W.W. and A.C. wrote the manuscript with input from all authors.

## Additional information

**Competing financial interests:** The authors declare no competing financial interests.

