## [Peer Review File · Nature Communications]

Reviewer #1 (Remarks to the Author)

Meddens et al. utilize the elegant imaging-analysis-technique STICS to investigate the complex organization dynamics of podosomes in dendritic cells.

The authors find that talin, vinculin, and F-actin exhibit flow patterns throughout podosome-rich regions within cells. Analysis of the data is solid and the results are interesting for the broad readership of Nature Comm. I have however a couple of concerns regarding the execution and presentation of the experiments. I suggest that the authors address these prior considerations for publication.

Major points:

(1) LifeAct is not an appropriate marker to report on changes in actin flow. Monomeric actin needs to be observed in order to gain quantitative understanding of the actin dynamics. I agree with the authors that LifeAct diffusion is much faster than the reported actin flow rates and should not affect the reported velocities as reported in Fig.1. However, the limiting factor is the association/dissociation kinetics of LifeAct which can be expected to be on the same order of magnitude as actin turnover and thus actin flow. This might or might not affect the velocities of 450nm/min as reported in Fig.S3 but definitely the velocities of 100nm/min as reported in Fig.8.

Consequently, comparing LifeAct-RFP and actin-RFP at one condition is not sufficient to support the author's claims. How do these plots actually prove that the conditions are not different? Fig. S3 reports 20% difference in the velocity magnitude.

To support the authors claims, they could perform FRAP experiments or a similar technique to quantify LifeAct kinetics and compare it to actin flow for the various conditions. Much more convincing would be to see results using monomeric actin as a reporter instead of LifeAct.

(2) In the manuscript, I did not find it clear if the authors examined fluorescently tagged myosin-II in green or an other color in the experiments in the presence of blebbistatin. The author should keep in mind that blebbistatin is sensitive to the 488nm laser light which deactivates blebbistatin. The authors report however that they see significant differences and repeat the experiments with the RHO/Rock kinase inhibitor Y27632. Can the author please clarify the blebbistatin experiment?

(3) The materials and methods section suggest that the statistics are low. Some of the experiments were only be performed one time such as the blebbistatin experiment. This is not sufficient considering the importance of the myosin-II controls as outlined above.

Minor points:

(1) Self-organization is a well-defined principle in cell-biology and the authors do not show any proof for the occurrence of self-organized mechanisms in their experiments. In fact, actin waves are predominately considered as self-assembled patterns rather than being self-organized. Can the authors please clarify and/or adjust the respective parts in the manuscript?

(2) The reported concentrations of the pharmacological treatments in the results and materials sections seem to differ. Can the authors clarify the final concentrations used in the experiments?

(3) Can the authors please use the same scaling for the velocity fields presented in Fig.S3?

(4) I found this statement in the abstract 'Furthermore, nanoscopy reveals myosin IIA-decorated actin filaments interconnecting multiple proximal podosomes.

Extending well-beyond podosome nearest neighbours, these actomyosin-dependent dynamic spatial patterns reveal a previously unappreciated mesoscale connectivity throughout the

podosome clusters.' confusing. The authors use nanoscale experiments but find mesoscopic changes. Can the authors please be more specific?

(5) The authors use the term "nanoscopy" in several occasions, specifically already in the abstract. I do not know whether this is already a well-enough perceived notation of super-resolution microscopy.

(6) How do the authors know in all of the figures that really podosomes are being observed? I know vinculin is a good marker, but it is not used as a control in all of the images.

Reviewer #2 (Remarks to the Author)

NCOMMS-16-07930A-Z Meddens et al.

The manuscript describes the movement and redistribution of podosome components which accounts for the controlled and synchronised podosome dynamics. The data show a mesoscale coordination in space and time of F-actin and actin binding protein dynamics ruled by actin polymerization and myosin IIA contractility. The main finding provided by the twSTICS and PVC analyses is that directional spatiotemporal coordination of cytoskeletal components exists between podosomes at larger spatial scales than previously reported (Poincloux et al.) i.e., extends beyond nearest neighbours and spans multiple proximal podosomes. The data also now show that this coordination reaches temporal scales up to 15 min. The authors suggest that such a mechanism may act as a platform for a continuous feedback loop for the cell's local protrusive activity, as required in front-rear polarity, migration and substrate degradation. Most results are obtained using a sophisticated technique based on the integration of nanoscopy images of the podosome cluster actomyosin machinery with measurements of the spatiotemporal molecular dynamics of podosome cytoskeletal components obtained with an extension of Spatiotemporal Image Correlation Spectroscopy for short time windows (twSTICS).

General comments and main concerns

- Many of the data obtained in the present manuscript rely on the use of an improved (extension) STICS technique described previously (Hebert et al., Brown et al.). The results concerning the podosome cluster mesoscale connectivity and correlated flow patterns of podosome components are novel and interesting. Altogether these data do support the conclusions that the podosome cluster represents a structured intracellular space, where self-organizing dynamic spatial patterns of podosome components occur within the cluster area, that guaranties the maintenance of the podosome cluster integrity and coordination over time.

- However, in its current presentation, the manuscript gives the impression that the authors have overexploited the technique leading to a loss of focus on the main objectives. Side experiments have been included in the manuscript at the expense of in-depth analysis of the essential aspects. On the one hand, the analysis of WASp movement appear superficial, not connected to the other results and thus do not add any value to the manuscript. On the other hand, some of the results seem too preliminary and would need further investigation. For instance, all the mechanistic aspects have been analysed only by using inhibitors while a complementary approach based on the use of an siRNA strategy would have strengthened the conclusions based on the pharmacological approach. More specifically, the exploration of the role of Myosin IIA in the coordination of the redistribution of cytoskeletal components throughout the cluster would benefit from this dual approach.

- Data presented in Fig.1 and movies 1 and 2, illustrate the plasticity of the podosome clusters in terms of spatial arrangements in DCs seeded on flat versus scratched surfaces. The scratched surface is an interesting experimental setup where mechanosensitive podosomes are challenged by the irregularities of the scratched glass. Podosome adhesive properties and podosome half-life

may differ on flat versus scratched surfaces. All subsequent experiments have been performed on flat glass only. It would be interesting to perform at least one other experiment on this surface, for instance the de novo podosome assembly in the nocodazole washout experiment, to examine podosome reformation and associated protein flow with this setup and determine if the scenario becomes more complex when adjacent podosomes are exposed to topologically distinct environments.

- Some experiments are difficult to follow: Some information is incomplete or missing regarding the experimental setup and/or statistical analysis of the data and/or lack of thorough description/annotation of the datasets presented in the figures.

- In many experiments, only a few cells were analysed (Fig 1, suppl. Fig 3, Fig.4d,g: 1-2 cells analysed).

Specific comments

- In the introduction it is stated "Understanding how cells mechanically explore their surroundings is an area of intense investigation". The present manuscript describes the spatiotemporal organisation of podosome clustering which constitutes the basis of the operating mechanism of a novel mechanosensor organelle. Filopodia are well known mechanosensor organelles which are not mentioned in the introduction.

- Fig 1e and f: These panels show the kymographs of Lifeact-RFP or Vinculin-GFP transfected cells. A cotransfection experiment would be more informative with joint acquisition of the data within the same cell (DC co-transfection is feasible according to data shown in Fig.5a), by using the same ROI, to follow molecular displacement and detection of a possible wave overlap. From the current kymographs it seems that Lifeact and vinculin waves are rather different. Could the authors comment on this approach?

- The same comment applies to Fig 1C. The analysis could have been done in one cell and using a single ROI used for the simultaneous analysis of the two proteins.

- Fig 2. At this point, it cannot be concluded that «podosome components are constantly redistributed in a spatially regulated manner within and throughout the podosome cluster, but that actin and vinculin are constantly redistributed in a spatially regulated manner within and throughout the podosome cluster

- Fig. 2b: the red line in the circle and the red line in the graph are not identical colors.

- Fig 4: Annoting Fig.4 would help to interpret the images and decipher the figure, ie., Lifeact-GFP(4a-d) and Vinculin-GFP (4e-h). Likewise, the nocodazole washout could be indicated on the figure as well as in movie 9. How many cells were analysed for the nocodazole washout experiment? (reappearance of the podosome clusters). For vinculin-GFP panel (4e) the time goes up to 20 min but in the corresponding kymograph (4f), the time indicated is 16 min. Fig. 4g is likely showing one representative experiment, The number of experiments and cells analysed are not indicated.

- Fig. 6a: The legend is not clear. Is it WASp-GFP that is shown in the image?

- Graph 6c: The result from this experiment are not clear. The fraction of empty vectors with WASp staining is smaller? What is the rationale for this comparison? Is it the mean fraction of ROIs that is shown? How many experiments have been performed? How many cells have been analysed? Error bars and statistics are missing.

- Fig. 7f,h - is it the mean that is shown? How many experiments have been performed? how many cells have been analysed- error bars and statistics are missing. Line 309 in the text refers to fig 7g, not 8g as stated. The statement « after CytoD talin vectors are fewer and the speeds slower (lines 310-311) is incorrect or misleading, there are more vectors with slower speeds according to the graph shown in Fig. 7i. These results should be clarified.

- Fig. 7 legend: "DCs adhering adherent..." (line 897)? Please correct the sentence. Panel a: both untreated and CytoD, phalloidin staining (could be indicated directly on the panel) Panels b-e: cells were cotransfected with Lifeact and talin. Altogether, the legend is unclear.

- Please state that Latrunculin (line 296) is an inhibitor of actin polymerisation and blebbistatin

(line 333) an inhibitor of myosin II.

- Myosin IIA - Blebbistatin is not a specific inhibitor of Myosin IIA. Myosin IIB is also sensitive to this inhibitor and some other myosins as well. Do DCs express Myosin IIB?. The images obtained when using the other inhibitors could be shown in the supplemental figures. ML7 and Y27632 inhibitors affect many other kinases beside MLC kinases. Please comment on these aspects.

- The authors have investigated the role of Myosin IIA activity (line 328). However, using blebbistatin it is difficult to conclude that the myosin IIA activity is involved as the drug is disrupting the podosome network and podosome cores seems to fuse together (Fig. 8b,c). Moreover, the intensity of the effect on velocity is difficult to appreciate as there are no indicators on the variability of the measures. It would be informative to know if the effect is weak or large. In fig 8f, are data from one cell per condition is plotted?

- Figure 8e: is it the mean that is shown? How many experiments have been performed? How many cells have been analysed? Error bars and statistics are missing.

Reviewer #3 (Remarks to the Author)

In Meddens et al., the authors applied their newly developed advanced quantitative image analysis methods (twSTICS), together with several microscopy techniques to dissect spatiotemporal dynamics of podosomes in dendritic cells. Mesoscale coordination between podosomes are demonstrated by correlated flow observed for actin and structural proteins such as talin and vinculin over large scale, and distinct from the more local core motion. Such correlated dynamics are determined to be dependent on the actin cytoskeletal integrity and myosin IIA contractility.

Overall, the strength and novelty of this manuscript is its meticulous application of advanced quantitative analysis to address the question on podosome dynamics. This reviewer is largely satisfied with the technical aspects of the experiments and analysis reported herein. Experiments reported in this study appears to be well-performed, with appropriate controls, and well-presented. However, a major criticism is that this reviewer finds there are still some rooms for improvement. For example, in the abstract the author stated that the study "provides mechanistic explanation of how podosome clusters function as coordinated mechanosensory area", while in the last sentence of the discussion the authors stated "Our study therefore provides a predictive model for mesoscale structural and dynamic organization of cytoskeletal structures testable in a variety of cell types." However, said "model" is described entirely in words in the text, and in a rather qualitative manner. It would greatly help if such mechanistic explanation is better articulated and illustrated, for example, in diagrams or some other visualization aid.

Also, in reference to the main hypothesis being tested in this study as stated in the introduction: "Here, we hypothesized that mesoscale dynamics exist that regulate movement and redistribution of structural components among podosomes to facilitate substrate sensing and efficient transmission of mechanical stimuli throughout the podosome cluster, possibly coordinating the cell's adhesive and protrusive activity." While the authors have convincingly demonstrated the existence of the mesoscale dynamics and the coordinated redistribution of the structural components among podosomes, this reviewer feels that the second part of the hypothesis: "to facilitate substrate sensing and efficient transmission of mechanical stimuli throughout the podosome cluster..." is not as well developed as it should be. With the techniques and system already developed, this reviewer feels that a few additional experiments may greatly improve this manuscript and more fully address the stated hypothesis. In particular, the relationship between the mesoscale dynamics and mechanosensing and their outcomes should be better established. For example, as shown in Fig. 1b, and the first section of Results, podosome formation appears to be highly sensitive to topographical pattern on the substrate. One possibility would be to investigate actin/talin/vinculin dynamics using twSTICS when cells are sensing such pattern. This should provide an interesting perturbation which would also be more relevant physiologically and mechanobiologically. In particular, it can be argued that the perturbation of podosome dynamics in this study as reported here using Cytochalasin D and Blebbistatin, while revealing actomyosin

requirements, may be a bit of a 'blunt' instrument since it is nearly tautological that once a structure is severely disrupted, it can be expected to be greatly impaired functionally.

In conclusion, the advanced methods and the findings reported here provide insights into an important question of how individual podosomes communicate with the others in the same cluster and function as a coordinated unit. The manuscripts may be greatly improved with a few additional experiments to strengthen the connection between the observed dynamics and mechanosensing by the cells, as suggested above.

Specific comments:

Concerning Fig7d and g in page 26: it is clearly shown that the velocity magnitude of LifeAct-RFP labeled actin network is substantially reduced after CytD treatment in the representative of the flow magnitude in Fig7b and paired comparison of quantification of flow magnitude in cells before and after adding CytD. However, the histogram of flow velocity of LifeAct-RFP after treated with CytD is virtually the same as that before CytD treatment. This seems surprising and should be explained better.

Minor comments:

Line 309: Fig. 8g should be Fig. 7g

Reviewer #1:

<Meddens et al. utilize the elegant imaging-analysis-technique STICS to investigate the complex organization dynamics of podosomes in dendritic cells. The authors find that talin, vinculin, and F-actin exhibit flow patterns throughout podosome-rich regions within cells. Analysis of the data is solid and the results are interesting for the broad readership of Nature Comm. I have however a couple of concerns regarding the execution and presentation of the experiments. I suggest that the authors address these prior considerations for publication.>

Major points:

<(I) LifeAct is not an appropriate marker to report on changes in actin flow. Monomeric actin needs to be observed in order to gain quantitative understanding of the actin dynamics. I agree with the authors that LifeAct diffusion is much faster than the reported actin flow rates and should not affect the reported velocities as reported in Fig.1. However, the limiting factor is the association/dissociation kinetics of LifeAct which can be expected to be on the same order of magnitude as actin turnover and thus actin flow. This might or might not affect the velocities of 450nm/min as reported in Fig.S3 but definitely the velocities of 100nm/min as reported in Fig.8. Consequently, comparing LifeAct-RFP and actin-RFP at one condition is not sufficient to support the author's claims. How do these plots actually prove that the conditions are not different? Fig. S3 reports 20% difference in the velocity magnitude. To support the authors claims, they could perform FRAP experiments or a similar technique to quantify LifeAct kinetics and compare it to actin flow for the various conditions. Much more convincing would be to see results using monomeric actin as a reporter instead of LifeAct.>

The reviewer raises an important point that we have carefully considered when we decided to use LifeAct-GFP to monitor and quantify the spatial patterns observed in dendritic cells. We agree with the reviewer that caution should be used when these FP-tagged probes are used in living cells. A number of publications are available where LifeAct-FP is being compared and contrasted to actin-FP and to other fluorescently tagged actin-binding proteins such as Utrophin and F-tractin (e.g. Riedl et al Nat Methods 2008; Sliogeryte et al J Biomech. 2016; Courtemanche et al Nat Cell Biol 2016). The overall consensus is that one should be careful with both GFP-tagged actin-binding probes and GFP-tagged monomeric actin and choose wisely depending on the process under investigation.

We can reassure the reviewer we have carefully considered the use of LifeAct-GFP for this study and we believe LifeAct-GFP, as used in our STICS approach, is an appropriate marker for the following reasons:

1) as suggested in Courtemanche et al Nat Cell Biol 2016, we carefully select cells expressing medium levels of LifeAct-GFP or LifeAct-RFP, to be sure that an exaggerated overexpression of LifeAct did not alter the actin structures. In fact, podosomes labelled with phalloidin after fixation and podosomes labelled with LifeAct-GFP or actin-GFP in living cells have exactly the same appearance: dotted core, radiating filaments, clustered organization. This was shown both in our previous publication (see van den Dries et al Nat Comms 2013) and in this study. We therefore are sure that the structures visible with LifeAct-GFP correspond to the endogenous podosomes.

2) the use of GFP-tagged monomeric actin instead of LifeAct, as suggested by the reviewer, is unfortunately not artefact-free either. It has been reported that formins –which we showed are enriched at podosomes and are important for preserving their actin architecture (see Panzer et al J Cell Sci 2016) - seem unable to incorporate GFP-tagged actin in polymerizing actin filaments (e.g. Chen et al J Struct Biol 2012). In addition, other studies reported that either actin-GFP would not be incorporated in all cytoskeletal structures (e.g.: Doyle & Botstein PNAS 1996; Wu & Pollard Science

2005; Yi, Wu *et al Mol Biol Cell* 2012) or would induce aberrations in cytoskeletal architecture and dynamics when highly overexpressed (e.g.: Westphal *et al Curr Biol* 1997; Aizawa *et al Cell Struct Funct* 1997). We noticed that the percentage of dendritic cells transfected with actin-GFP is always somehow lower than the percentage of dendritic cells transfected with LifeAct-GFP, indicating that these primary cells might find overexpression of monomeric actin toxic, whereas they have no problems in expressing LifeAct-FP.

3) we might have misled the reviewer by stating we were imaging *flow/flux* of actin. We did not mean to claim we used STICS to specifically look at actin polymerization flow. We used the word 'flow' to describe the directional spatial patterns observed for the fluorescence intensity of actin and actin-interacting proteins such as talin and vinculin within the podosome cluster. In fact, classical F-actin flow (i.e. polymerization rate) has been already calculated in several publications (e.g. Gardel *et al J Cell Biol* 2008) and is reported to be about 15-30 nm/s, which corresponds to 0.9-1.8 $\mu\text{m}/\text{min}$. This value is much higher than what we get from the STICS movies for LifeAct- and actin-GFP, which is about 0.10-0.15 $\mu\text{m}/\text{min}$ and is exactly in the same order of magnitude of the velocity measured for vinculin-GFP. The purpose of our study was purely to measure and understand the dynamic spatial patterns occurring throughout this mechanosensitive podosome cluster.

4) we have already demonstrated that podosomes labelled with LifeAct-GFP display protruding oscillations similar to actin-GFP labelled podosomes (see van den Dries *et al Nat Comms* 2013). Moreover, other groups have also used LifeAct to monitor podosome dynamics (e.g.: Labernadie *et al Nat Comms* 2014), indicating that this probe is widely accepted to monitor podosome dynamics.

We do however agree with the reviewer about the fact that the comparison between actin-GFP and LifeAct-GFP performances could be more robust. Therefore, to strengthen our statement that LifeAct-GFP and actin-GFP give similar STICS results, we have now recorded more movies of dendritic cells expressing either LifeAct-GFP or actin-GFP, we have analysed them by STICS, retrieved the average velocity values and performed the PVC analysis. These new results show average velocity values ranging between 0.10-0.15 $\mu\text{m}/\text{min}$ for both LifeAct-GFP and actin-GFP. We now added the scatter plot of the average velocity values as well as representative PVC plots for each protein to the **revised Suppl Fig 3**. These plots show that the average velocities as retrieved by STICS are in the same order of magnitude and that the spatiotemporal correlation among the vectors as visualized by PVC is similar for LifeAct-GFP and actin-GFP.

As requested by the reviewer, we have also performed FRAP at the podosome cluster in dendritic cells expressing either LifeAct-GFP or actin-GFP. As shown below in **confidential Figure #1**, FRAP shows the very quick recovery (fast diffusion) of LifeAct-GFP to the actin filaments and the slower recovery of the actin-GFP.

In our previous work, we did already perform FRAP of actin and other actin-binding proteins at individual podosomes under different conditions, i.e. CytoD and Blebb treatments, and found differences in their fast diffusion only after CytoD treatment (see van den Dries *et al Nat Comms* 2013). The FRAP approach provides different information with respect to STICS, therefore the FRAP based recovery time values cannot be compared with the vector velocity values obtained here by STICS. In addition, FRAP does not provide information on the directionality, which is in fact the key finding of our study.

CONFIDENTIAL FIG. 1. FRAP curves of LifeAct-GFP (Cell 1 and 2) and actin-GFP (Cell 3 and 4). These curves are representative of >10 cells per each condition.

We hope we have convincingly motivated our choice to use LifeAct-GFP to measure the dynamic spatial patterns observed in this study. Nevertheless, the reviewer’s comment prompted us to add a short text in the main manuscript (see **pag. 5**) to avoid confusion and emphasize that we are not talking about actin polymerization flow but rather we use the word ‘flow’ to indicate the directionality of the fluorescent signal displacement. Also, we have better described the comparison between LifeAct-GFP and actin-GFP (see **new legend of Suppl fig 3**).

<(2) In the manuscript, I did not find it clear if the authors examined fluorescently tagged myosin-II in green or another color in the experiments in the presence of blebbistatin. The author should keep in mind that blebbistatin is sensitive to the 488nm laser light which deactivates blebbistatin. The authors report however that they see significant differences and repeat the experiments with the RHO/Rock kinase inhibitor Y27632. Can the author please clarify the blebbistatin experiment?>

We regret we have not clearly described the blebbistatin experiments. We did not examine fluorescently tagged myosin IIA. Unfortunately, dendritic cells seem to refuse to overexpress myosin IIA, no matter which transfection trick we use. This prevents us from directly looking at myosin II dynamics within the podosome cluster. To clarify this part for the reader, we now clearly state that overexpression of myosin IIA is unsuccessful and prevents recording movies of its dynamics and that the SIM data are from fixed DCs (see **pag. 9**).

Regarding the 488nm laser light issue when using blebbistatin in living cells, we can reassure the reviewer we are absolutely aware of it. In fact, during live cell imaging we never use GFP-tagged proteins but rather RFP- or mCherry-tags when blebbistatin is added (see also our previous publication *van den Dries et al Nat Comms 2013*). We had already stated this in the original manuscript in the Materials and Methods section (see **“Live cell imaging” section, pag. 13**). We used blebbistatin in the SIM experiments with fixed cells (Fig 8a,b), where we labelled both actin and myosin IIA, and for the STICS analysis of living cells (Fig 8c-f), where we indeed strictly used RFP-tagged LifeAct. We used the ROCK inhibitor Y27632 and the MLC phosphatase inhibitor ML7 in SIM experiments of fixed cells (mentioned as data not shown in the original manuscript) and they

showed disturbed actin network similarly to what we observed when myosin IIA was blocked with blebbistatin (Fig 8b). Although we already showed in our previous publication that blocking myosin IIA with various drugs showed similar effects on podosome dynamic protrusions (*van den Dries et al Nat Comms 2013*), the reviewer's comment prompted us to include the SIM images of dendritic cells after blocking myosin IIA with ML7 and Y27632 (see **new Supplementary fig. 8c**).

We are confident these changes are now better describing our attempts to block myosin IIA activity and determine the effects on podosome mesoscale dynamics and architecture.

<(3) The materials and methods section suggest that the statistics are low. Some of the experiments were only be performed one time such as the blebbistatin experiment. This is not sufficient considering the importance of the myosin-II controls as outlined above.>

Some of the stainings of fixed DCs for SIM were indeed only performed once or twice because they were solely meant as supporting evidence to the novel STICS analysis. In fact, some of these very same stainings (+/- cytochalasin D, +/- blebbistatin) were already performed and imaged by standard confocal microscopy in previous work (*van den Dries et al Nat Comms 2013*). The novelty in the current manuscript is the SIM analysis that much better resolves the architecture of podosomes after these treatments and nicely visualizes myosin IIA molecules decorating the radiating actin filaments at an unprecedentedly high resolution. Furthermore, podosomes before and after myosin IIA block by Y27632 and ML7 were imaged by SIM (and in the past by confocal microscopy, see *van den Dries et al Nat Comms 2013*), showing a disturbed actin network integrity with strong decrease of colocalizing myosin staining similarly to the blebbistatin treated samples. As mentioned above, we now included these data as **new Suppl. Fig. 8c**, to further support the blebbistatin data.

Importantly, the novel STICS experiments after blebbistatin treatment concern >10 movies recorded over 3 different days, which gave very similar values of vector velocity (see **fig 8d**). These results together with the supporting SIM images as well as our previous published work concur in demonstrating the involvement of myosin IIA in mediating the dynamic spatial patterns of podosome components. We hope the reviewer will agree there is now sufficient evidence to support our claims.

Minor points:

<(1) Self-organization is a well-defined principle in cell-biology and the authors do not show any proof for the occurrence of self-organized mechanisms in their experiments. In fact, actin waves are predominately considered as self-assembled patterns rather than being self-organized. Can the authors please clarify and/or adjust the respective parts in the manuscript?>

Following the reviewer's suggestion, we have now used the term self-assembled instead of self-organized throughout the revised manuscript.

<(2) The reported concentrations of the pharmacological treatments in the results and materials sections seem to differ. Can the authors clarify the final concentrations used in the experiments?>

We apologize for our sloppiness. We have checked once again all our old movies and added the newly recorded ones. The correct concentration range of all pharmacological treatments are now consistently stated throughout the manuscript and the supplementary information.

<(3) Can the authors please use the same scaling for the velocity fields presented in Fig.S3?>

As suggested, we have changed the scaling (see revised Suppl Fig 3).

<(4) I found this statement in the abstract 'Furthermore, nanoscopy reveals myosin IIA-decorated actin filaments interconnecting multiple proximal podosomes. Extending well-beyond podosome nearest neighbours, these actomyosin-dependent dynamic spatial patterns reveal a previously unappreciated mesoscale connectivity throughout the podosome clusters.' confusing. The authors use nanoscale experiments but find mesoscopic changes. Can the authors please be more specific?>

We have slightly changed the sentence in the abstract to improve clarity and better explain that nanoscale imaging revealed actin structures connecting multiple neighboring podosomes and enriched in myosin IIA. This contractile actomyosin network is mediating the mesoscale waves as observed by STICS.

<(5) The authors use the term "nanoscopy" in several occasions, specifically already in the abstract. I do not know whether this is already a well-enough perceived notation of super-resolution microscopy.>

We can reassure the reviewer that the term nanoscopy is more and more widely used in scientific publications (e.g.: Chmyrov et al *Nat Meth* 2013; Hell S.W. *Science* 2007, Bourg et al *Nat Photonics* 2015), book titles (<https://www.crcpress.com/Optical-Nanoscopy-and-Novel-Microscopy-Techniques/Xi/p/book/9781466586291>; <https://www.crcpress.com/Cell-Membrane-Nanodomains-From-Biochemistry-to-Nanoscopy/Cambi-Lidke/p/book/9781482209891>) and conference titles (<http://www.icon-europe.org/>). If the reviewer agrees, we would like to keep this term in the manuscript.

<(6) How do the authors know in all of the figures that really podosomes are being observed? I know vinculin is a good marker, but it is not used as a control in all of the images.>

Podosomes have a very distinct shape, clustered organization and typical 'pulsating' activity (in living cells). There are even image analysis algorithms (from us and the Linder's laboratory) that are uniquely based on the actin signal and used to count podosome numbers, distances, etc. Co-staining of vinculin in dendritic cells is not really necessary to confirm the features are podosomes.

Reviewer #2 (Remarks to the Author):

<The manuscript describes the movement and redistribution of podosome components which accounts for the controlled and synchronised podosome dynamics. The main finding provided by the twSTICS and PVC analyses is that directional spatiotemporal coordination of cytoskeletal components exists between podosomes at larger spatial scales than previously reported (Poincloux et al.) i.e., extends beyond nearest neighbours and spans multiple proximal podosomes. Most results are obtained using a sophisticated technique based on the integration of nanoscopy images of the podosome cluster actomyosin machinery with measurements of the spatiotemporal molecular dynamics of podosome cytoskeletal components obtained with an extension of Spatiotemporal Image Correlation Spectroscopy for short time windows (twSTICS).>

General comments and main concerns

<- Many of the data obtained in the present manuscript rely on the use of an improved (extension) STICS technique described previously (Hebert et al., Brown et al.). The results concerning the podosome cluster mesoscale connectivity and correlated flow patterns of podosome components are novel and interesting. Altogether these data do support the conclusions that the podosome cluster represents a structured intracellular space, where self-organizing dynamic spatial

patterns of podosome components occur within the cluster area, that guaranties the maintenance of the podosome cluster integrity and coordination over time.>

1) <However, in its current presentation, the manuscript gives the impression that the authors have overexploited the technique leading to a loss of focus on the main objectives. Side experiments have been included in the manuscript at the expense of in-depth analysis of the essential aspects. On the one hand, the analysis of WASp movement appear superficial, not connected to the other results and thus do not add any value to the manuscript. On the other hand, some of the results seem too preliminary and would need further investigation. For instance, all the mechanistic aspects have been analysed only by using inhibitors while a complementary approach based on the use of an siRNA strategy would have strengthened the conclusions based on the pharmacological approach. More specifically, the exploration of the role of Myosin IIA in the coordination of the redistribution of cytoskeletal components throughout the cluster would benefit from this dual approach.>

We thank the reviewer for his/her constructive comments and suggestions. Below, we will **1)** motivate our choice to include the WASP experiments and **2)** explain the technical issues that prevent extensive manipulation of these primary cells.

(1) The WASP experiments were performed to show to what extent the podosome core movements (sliding, fusion, fission) would contribute to the dynamic spatial patterns revealed by STICS. Due to the complex actin arrangements present at podosomes (i.e. branched core, inter-podosome cables and podosome-membrane connecting filaments), we wanted to understand whether all these actin structures contributed to the dynamic spatial patterns. With its specific localization at the center of the actin core, WASP-GFP is a perfect probe to specifically monitor the core movements and its use revealed a modest contribution of core sliding, fusing and splitting to the flow patterns (former main fig 6). We find these data useful to understand which factors contribute to the dynamic spatial pattern and would like to keep them in the manuscript. However, we understand the reviewer's remark and moved the WASP data to the supplementary information file, as we think they might be better suited as supplementary figure (**see new Suppl Fig. 6**). The main text has been accordingly adjusted as well.

(2) We would love to be able to perform RNAi or siRNA in dendritic cells and monitor dynamics of FP-tagged cytoskeletal proteins, however these are human primary cells, fully differentiated from fresh blood monocytes of healthy donors, which have a limited life span in culture (8-10 days) and are pretty difficult to manipulate with respect to a cell line. We can perform RNAi/siRNA to some extent (although sometimes whether the KO is successful very much depends on the targeted protein itself) and we can transfect plasmids encoding for fluorescently tagged proteins (although also here the yield is not higher than 10-20%). However, it is really impossible to do both transfection and RNAi in the same cells. This is why we have to rely on pharmacological treatments. In addition, RNAi of myosin IIA would most likely result in reduced/no podosome formation, thus preventing any dynamic study, as basal myosin activity is necessary for podosome formation and maintenance (*Berdeaux et al., J Cell Biol 2004; Kopp et al., Mol Biol Cell 2006; Thatcher et al., J Pharmacol Sci 2011*). Furthermore, we previously showed that whereas basal myosin activity cooperates with actin to drive podosome protruding oscillations (*van den Dries et al Nat Comms 2013*), enhanced myosin IIA-mediated contraction, induced by prostaglandin E2 via Rho-Rho kinase activation, leads to immediate podosome dissolution (*van Helden et al., J Cell Sci 2008, van den Dries et al., Cell Mol Life Sci 2012*). This indicates that a delicate signalling threshold must exist to regulate myosin IIA contractility and subsequently modulate podosome formation, dissolution and protruding activity. Here, we use both pharmacological treatments and super-

resolution imaging to respectively determine the role of myosin IIA in the dynamic spatial patterns of podosome components and to visualize the specific localization of endogenous myosin IIA molecules at the actin filaments radiating from the podosome cores.

Therefore, taking all our past and new results together, we hope we have convinced the reviewer that the data on myosin IIA presented in this manuscript are not ‘too preliminary’.

2) <Data presented in Fig.1 and movies 1 and 2, illustrate the plasticity of the podosome clusters in terms of spatial arrangements in DCs seeded on flat versus scratched surfaces. The scratched surface is an interesting experimental setup where mechanosensitive podosomes are challenged by the irregularities of the scratched glass. Podosome adhesive properties and podosome half-life may differ on flat versus scratched surfaces. All subsequent experiments have been performed on flat glass only. It would be interesting to perform at least one other experiment on this surface, for instance the de novo podosome assembly in the nocodazole washout experiment, to examine podosome reformation and associated protein flow with this setup and determine if the scenario becomes more complex when adjacent podosomes are exposed to topologically distinct environments.>

As pointed out by the reviewer, the scratched surfaces are indeed a handy setup to challenge podosome organization. As shown in our previous publication by confocal microscopy (*van den Dries et al Cell Mol Life Sci 2012*) and in current **Suppl Fig 1** by SIM, podosomes nicely align along ridges or scratches, respectively. We did not observe any significant change in the composition (actin and vinculin) of the podosomes on flat glass with respect to scratched glass or ridges (**see Supplementary Fig 1**), which we interpreted as no differences in adhesive properties. However, considering the reviewer’s remark, we have imaged myosin IIA and actin on dendritic cells adhering onto scratched surfaces and now display these results as **new Supplementary Fig. 8**. Since the presence of actin, myosin and vinculin is pretty similar on flat and scratched glass, we assume that the podosome main molecular constituents and overall podosome organization are conserved.

As suggested by the reviewer, we have performed new time-lapse movies of dendritic cells expressing LifeAct-GFP and adhering onto scratched glass coverslips. Interestingly, in the scratches we often see elongated podosome cores that dynamically tend to almost fuse together moving (almost sliding) along the linear topographical cue, often fusing and splitting. From these movies, we have measured (1) protruding oscillations comparing podosomes on the scratch with neighbouring podosomes on flat areas and (2) dynamic spatial patterns of existing as well as de novo formed podosome clusters (after nocodazole washout) on scratches (**see new main Fig. 5, supplementary movies 11 and 12**).

(1) similarly to Fig 1, we measured the fluctuations of actin fluorescence on neighbouring podosomes that were visually connected but experienced different topography, i.e. one was on a scratch and its neighbour on the flat surface. Independently of the topography sensed, proximal podosomes show correlated vertical oscillations (**see new Fig. 5c**), indicating that sensing irregular topography does not disturb the overall inter-podosome structural communication.

(2) twSTICS and subsequent PVC analysis interestingly reveal how the podosome cluster globally responds to the topographical cues encountered. More specifically we now demonstrate that the presence of the scratches clearly promotes directionality of cluster movement as demonstrated by a highly increased spatiotemporal correlation among all the vectors within a cluster (**Fig. 5d,e and supplementary movie 11**). Moreover, the scratches represent nucleation sites for podosome reformation, as their presence seems to accelerate spontaneous de novo podosome assembly after

nocodazole addition and washout (**Fig. 5f-i and supplementary movie 12**). At **page 7** of the revised main manuscript, we now added an entirely new paragraph to describe these new results.

Unfortunately we cannot apply twSTICS to “*determine if the scenario becomes more complex when adjacent podosomes are exposed to topologically distinct environments*” as we cannot distinguish between the dynamic spatial patterns of two individual, proximal podosomes. The ROI used for the STICS analysis is 2.1x2.1 μm (see Supplementary fig 2), which encompasses a few neighbouring podosomes (each with an average diameter of 500 nm) and provides one vector. In addition, neighbouring vectors derive from two partially overlapping ROIs (75% overlapping), as explained in the Materials and Methods. It is therefore technically impossible to compare the individual STICS output of two proximal podosomes with sufficient spatial sampling for STICS. Decreasing the size of the ROI is not a good option either because the STICS correlation functions become too noisy to fit the peak and obtain a reliable velocity. We realize this might be an interesting information for the reader so we added a short sentence in the **Supplementary Text (page 10)**.

We believe the new experiments using scratched surfaces now provide an important conceptual advancement that indeed strengthens the notion that the podosome cluster is a mechanosensing and mechanotransducing platform in dendritic cells.

3) *<Some experiments are difficult to follow: Some information is incomplete or missing regarding the experimental setup and/or statistical analysis of the data and/or lack of thorough description/annotation of the datasets presented in the figures.>*

To comply with the journal space limitations, some information had been pooled and only provided in Materials and Methods. However, we carefully went through the entire manuscript and have revised Materials & Methods as well as all figure and movie legends.

4) *<In many experiments, only a few cells were analysed (Fig 1, suppl. Fig 3, Fig.4d,g: 1-2 cells analysed).>*

We have performed new experiments to strengthen the statistics related to the figures mentioned by the reviewer. The number of cells imaged and the number of experiments are provided in the revised Materials and Methods and we did an effort to merge two representative movies for each condition to indicate the reproducibility of our findings.

Specific comments

5) *<In the introduction it is stated "Understanding how cells mechanically explore their surroundings is an area of intense investigation". The present manuscript describes the spatiotemporal organisation of podosome clustering which constitutes the basis of the operating mechanism of a novel mechanosensor organelle. Filopodia are well known mechanosensor organelles which are not mentioned in the introduction.>*

We thank the reviewer for this suggestion. We have now mentioned the filopodia and refer to a recent review (*Jacquemet et al Curr Opin Cell Biol 2015*) that discuss the role of filopodia in substrate sensing and cell mechanics.

6) <Fig 1e and f: These panels show the kymographs of Lifeact-RFP or Vinculin-GFP transfected cells. A cotransfection experiment would be more informative with joint acquisition of the data within the same cell (DC cotransfection is feasible according to data shown in Fig.5a), by using the same ROI, to follow molecular displacement and detection of a possible wave overlap. From the current kymographs it seems that Lifeact and vinculin waves are rather different. Could the authors comment on this approach?>

7) <The same comment applies to Fig 1C. The analysis could have been done in one cell and using a single ROI used for the simultaneous analysis of the two proteins.>

Since both comment 6 and 7 concern Figure 1, we address them together.

Figure 1e,f shows indeed separate kymographs of actin and vinculin as we wanted to build up this manuscript gradually. Showing correlated kymographs already in Fig 1 would give away the main message of the paper, i.e. the existence of correlated dynamic spatial patterns of cytoskeletal molecules that travel throughout the podosome cluster. The same applies for Fig 1c, which we believe will be clearer for the reader if we display three examples of podosome pairs and keep the plots of LifeAct and vinculin separate.

Regarding the waves of actin and vinculin being different in the current kymographs of Fig 1e,f, one should consider that these waves are anyway highly irregular events. Both within the same cell and between two cells waves display differences in spatiotemporal coordination, which is why the PVC analysis turned out to be so useful to quantify in one plot all vector correlations from one movie. Moreover, the LifeAct movies also show the actin cores, not only the actin network (where vinculin molecules are located, see also *van den Dries et al Mol Biol Cell 2013*), meaning that the actin mesoscale dynamics might appear slightly different in a kymograph with respect to vinculin. However, we show that vinculin and actin, talin and actin (Fig. 6), vinculin and talin (Supplementary Fig 5) exhibit directionally correlated flows, meaning that there is a high degree of wave overlapping.

We hope the reviewer will agree with us to keep Figure 1 in its current form.

8) <Fig 2. At this point, it cannot be concluded that «podosome components are constantly redistributed in a spatially regulated manner within and throughout the podosome cluster, but that actin and vinculin are constantly redistributed in a spatially regulated manner within and throughout the podosome cluster»>

As suggested by the reviewer, we corrected our statement (see **pag. 5, revised manuscript**).

9) <Fig. 2b: the red line in the circle and the red line in the graph are not identical colors.>

We have corrected the color.

10) < Fig 4: Annoting Fig.4 would help to interpret the images and decipher the figure, ie., Lifeact-GFP (4a-d) and Vinculin-GFP (4e-h). Likewise, the nocodazole washout could be indicated on the figure as well as in movie 9. How many cells were analysed for the nocodazole washout experiment? (reappearance of the podosome clusters). For vinculin-GFP panel (4e) the time goes up to 20 min but in the corresponding kymograph (4f), the time indicated is 16 min. Fig. 4g is likely showing one representative experiment, The number of experiments and cells analysed are not indicated.>

The washout starts already on the first frame of the movie, so we slightly changed the legend so that this figure is clearer. We have added the requested information throughout the revised manuscript and we corrected the 16 min to 20 min (it was a sloppy copy-paste mistake from the kymograph above). To respect the journal space constraints, we have pooled the information regarding number of experiments/cells in the Materials and Methods sections. In any case, we have recorded spontaneously moving clusters, like the one shown in Fig. 4g, in several cells and we now show two representative movies for this phenomenon (see **new supplementary movie 10**).

11) <Fig. 6a: The legend is not clear. Is it WASp-GFP that is shown in the image? Graph 6c: The result from this experiment are not clear. The fraction of empty vectors with WASp staining is smaller? What is the rationale for this comparison? Is it the mean fraction of ROIs that is shown? How many experiments have been performed? How many cells have been analysed? Error bars and statistics are missing.>

We apologize for not having been clear enough. Fig 6, which is now moved to the Supplementary Information file as the **new Suppl Fig. 6**, indeed shows WASp-GFP, as mentioned in the legend. The legend did contain statistic test used but we agree it should be adjusted to improve clarity. Panel c and d are supposed to dissect the difference shown in panel b, where WASP-GFP displays a lower average velocity than LifeAct-GFP. Specifically panel c shows no significant difference in the number of empty ROIs (plot is obtained by adding all vectors retrieved from all movies available), indicating that the slower average velocity of WASP is due to a higher number of vectors with lower velocity values. We have now added the missing information, including number of experiments, to the legend of **new Suppl fig 6**.

12) <Fig. 7f,h - is it the mean that is shown? How many experiments have been performed? how many cells have been analysed- error bars and statistics are missing. Line 309 in the text refers to fig 7g, not 8g as stated. The statement « after CytoD talin vectors are fewer and the speeds slower (lines 310-311) is incorrect or misleading, there are more vectors with slower speeds according to the graph shown in Fig. 7i. These results should be clarified. Fig. 7 legend: "DCs adhering adherent..." (line 897)? Please correct the sentence. Panel a: both untreated and CytoD, phalloidin staining (could be indicated directly on the panel) Panels b-e: cells were cotransfected with Lifeact and talin. Altogether, the legend is unclear.>

We thank the reviewer for his/her suggestions and apologize for our sloppiness. Panel **f,h** were not mean values but a pool of all the vectors from all the cells analyzed, which explains why no error bars were shown, however we should have explained this better. For better clarity, we now display scatter plots for panels **f** and **h** to better indicate data spreading among the various cells. We have now carefully revised the order of the panels and improved the legend (**see revised Fig. 7**).

13) < Please state that Latrunculin (line 296) is an inhibitor of actin polymerisation and blebbistatin (line 333) an inhibitor of myosin II.>

We added both statements.

14) <Myosin IIA - Blebbistatin is not a specific inhibitor of Myosin IIA. Myosin IIB is also sensitive to this inhibitor and some other myosins as well. Do DCs express Myosin IIB?. The images obtained when using the other inhibitors could be shown in the supplemental figures. ML7 and Y27632 inhibitors affect many other kinases beside MLC kinases. Please comment on these aspects.>

In a previous publication (see *van Helden et al J Cell Sci 2008*) it was already reported by Western Blot that DCs express only myosin IIA, as no myosin IIB or IIC could be detected. As suggested by the reviewer, we now show the SIM images of the actin and myosin IIA after ML7+Y27632 combined treatment (see **new supplementary fig. 8c**). We understand the reviewer's concern, no inhibitor is ever completely specific. However, blebbistatin, ML7 and Y27632 are widely used to block myosin II activity, albeit through different mechanisms. We now shortly discuss these aspects **at page 9 of the revised manuscript**.

15) <The authors have investigated the role of Myosin IIA activity (line 328). However, using blebbistatin it is difficult to conclude that the myosin IIA activity is involved as the drug is disrupting the podosome network and podosome cores seems to fuse together (Fig. 8b,c). Moreover, the intensity of the effect on velocity is difficult to appreciate as there are no indicators on the variability of the measures. It would be informative to know if the effect is weak or large. In fig 8f, are data from one cell per condition is plotted? >

We have blocked myosin IIA by blebbistatin as well as ML7+Y27632. SIM images here as well as our previous work show that the podosome cluster is altered but not completely dissolved. We indeed previously showed that labelling of podosome markers such as talin, vinculin, paxillin and zyxin was maintained even after blebbistatin treatment, although the vertical oscillations were abolished (*van den Dries et al Nat Comms 2013*). Here, the SIM data of Fig 8a,b, now supported by the new suppl fig 8, confirm that myosin IIA crosslinks all podosomes within a cluster and that its block specifically reduces the network integrity. As a consequence, average velocities values are lower and the fraction of empty ROIs is higher. To better show the variability of the measurements, we now show scatter plots instead of histogram plots to depict the average velocities as well as the fraction of ROIs without vectors (see **revised fig. 8**). In addition, as **confidential figure #2** for the reviewer, we here show a plot where the average mean velocity values of the various proteins are displayed to give an idea of the data spreading.

CONFIDENTIAL FIGURE 2. Overview of the average velocity values calculated for LifeAct, vinculin and talin. Each dot is the average value from individual cells from several movies recorded over multiple experiments.

In the past we reported that during blebbistatin treatment the core became larger as the core branched actin pool kept polymerizing for a while (see *van den Dries et al Nat Comms 2013*), which could appear as if the cores are fused together.

We hope the reviewer will agree with us that by including the SIM images after myosin IIA block by ML7+Y27632, which fully support the blebbistatin data, and together with published work and the effect of myosin IIA block on the STICS output, we have provided sufficient evidence for a role of myosin IIA in the podosome cluster mesoscale dynamics.

16) <- Figure 8e: is it the mean that is shown? How many experiments have been performed? How many cells have been analysed? Error bars and statistics are missing.>

We have updated all plots showing ‘Fraction of empty ROIs’ like old Figure 8e. They are now displayed as scatter plots with each dot representing a single cell and significance testing has been performed as described in the figure legends. As already mentioned above, we had included the information regarding cell numbers in the Materials and Methods. However, we do realize this may not be sufficiently visible, so we have included these numbers directly in the legends.

Reviewer #3 (Remarks to the Author):

<In Meddens et al., the authors applied their newly developed advanced quantitative image analysis methods (twSTICS), together with several microscopy techniques to dissect spatiotemporal dynamics of podosomes in dendritic cells. Mesoscale coordination between podosomes are demonstrated by correlated flow observed for actin and structural proteins such as talin and vinculin over large scale, and distinct from the more local core motion. Such correlated dynamics are determined to be dependent on the actin cytoskeletal integrity and myosin IIA contractility.>

1) <Overall, the strength and novelty of this manuscript is its meticulous application of advanced quantitative analysis to address the question on podosome dynamics. This reviewer is largely satisfied with the technical aspects of the experiments and analysis reported herein. Experiments reported in this study appears to be well-performed, with appropriate controls, and well-presented. However, a major criticism is that this reviewer finds there are still some rooms for improvement. For example, in the abstract the author stated that the study "provides mechanistic explanation of how podosome clusters function as coordinated mechanosensory area", while in the last sentence of the discussion the authors stated "Our study therefore provides a predictive model for mesoscale structural and dynamic organization of cytoskeletal structures testable in a variety of cell types." However, said "model" is described entirely in words in the text, and in a rather qualitative manner. It would greatly help if such mechanistic explanation is better articulated and illustrated, for example, in diagrams or some other visualization aid. >

We are grateful to the reviewer for the positive comments and constructive suggestions. We agree with the reviewer that a visualization of the studied phenomena would help the readership to better grasp the concept of traveling waves of cytoskeletal components. We have therefore prepared an animation as new **supplementary movie 21** with a corresponding figure depicting snapshots of the molecule fluxes observed within a podosome cluster (see **new fig 9** and **new supplementary movie 21**). For the sake of simplicity, we only show inter-podosome connecting filaments. We hope the reviewer will find our animation sufficiently explanatory.

2) <Also, in reference to the main hypothesis being tested in this study as stated in the introduction: "Here, we hypothesized that mesoscale dynamics exist that regulate movement and redistribution of structural components among podosomes to facilitate substrate sensing and efficient transmission of mechanical stimuli throughout the podosome cluster, possibly coordinating the cell's adhesive and protrusive activity." While the authors have convincingly demonstrated the existence of the mesoscale dynamics and the coordinated redistribution of the structural components among podosomes, this reviewer feels that the second part of the hypothesis: "to facilitate substrate sensing and efficient transmission of mechanical stimuli throughout the podosome cluster..." is not as well developed as it should be. With the techniques and system already developed, this reviewer feels that a few additional experiments may greatly improve this manuscript and more fully address the stated hypothesis.

In particular, the relationship between the mesoscale dynamics and mechanosensing and their outcomes should be better established. For example, as shown in Fig. 1b, and the first section of Results, podosome formation appears to be highly sensitive to topographical pattern on the substrate. One possibility would be to investigate actin/talin/vinculin dynamics using twSTICS when cells are sensing such pattern. This should provide an interesting perturbation which would also be more relevant physiologically and mechanobiologically. In particular, it can be argued that the perturbation of podosome dynamics in this study as reported here using Cytochalasin D and Blebistatin, while revealing actomyosin requirements, may be a bit of a 'blunt' instrument since it is nearly tautological that once a structure is severely disrupted, it can be expected to be greatly impaired functionally. In conclusion, the advanced methods and the findings reported here provide insights into an important question of how individual podosomes communicate with the others in the same cluster and function as a coordinated unit. The manuscripts may be greatly improved with a few additional experiments to strengthen the connection between the observed dynamics and mechanosensing by the cells, as suggested above.>

We thank the reviewer for his/her constructive comments. As suggested (and in line with comment nr 2 of reviewer #2), we have performed additional experiments using the scratched glass surfaces.

First, we have imaged myosin IIA and actin on dendritic cells adhering onto scratched surfaces and now display these results as **new Supplementary Fig. 8b**. Since the presence of actin, myosin

and vinculin is pretty similar on flat and scratched glass, we assume that the podosome main molecular constituents and overall organization of individual podosomes are conserved independently from the topography encountered.

Second, we have performed new time-lapse movies of dendritic cells expressing LifeAct-GFP and adhering onto scratched glass coverslips. Interestingly, in the scratches we often see elongated podosome cores that dynamically tend to almost fuse together moving (almost sliding) along the linear topographical cue, often fusing and splitting. From these movies, we have measured **(1)** protruding oscillations comparing podosomes on the scratch with neighbouring podosomes on flat areas and **(2)** dynamic spatial patterns of existing as well as de novo formed podosome clusters (after nocodazole washout) on scratches (see new main Fig. 5, supplementary movies 11 and 12).

(1) similarly to Fig 1, we measured the fluctuations of actin fluorescence on neighbouring podosomes that were visually connected but experienced different topography, i.e. one was on a scratch and its neighbour on the flat surface. Independently of the topography sensed, proximal podosomes show correlated vertical oscillations (see new Fig. 5c), indicating that sensing irregular topography does not disturb the overall inter-podosome structural communication.

(2) twSTICS and subsequent PVC analysis interestingly reveal how the podosome cluster globally responds to the topographical cues encountered. More specifically we now demonstrate that the presence of the scratches clearly promotes directionality of cluster movement as demonstrated by a highly increased spatiotemporal correlation among all the vectors within a cluster (Fig. 5d,e and supplementary movie 11). Moreover, the scratches represent nucleation sites for podosome reformation, as their presence seems to accelerate spontaneous de novo podosome assembly after nocodazole addition and washout (Fig. 5f-i and supplementary movie 12). At page 7 of the revised main manuscript, we now added an entirely new paragraph to describe these new results.

Regarding the CytD and Blebbistatin perturbations, we of course agree with the reviewer that ideally one would like to have an inhibitor that specifically blocks dynamics without affecting structure. Unfortunately, the intimate relationship between structure and function together with the enormous amount of molecular components involved make sure that almost no function block is possible without affecting the architecture of podosomes. These multimolecular assemblies are constantly remodelled both at the individual podosome and at the collective level, as they respond to the substrate and to their neighbours. Considering also the primary nature of the dendritic cells, which prevents more sophisticated perturbations such as knock-out and knock-in etc, the use of soluble inhibitors is sometimes the only approach available.

We hope the reviewer will agree that despite their limitations, the pharmacological perturbations performed here, supported by previous work and images of fixed cells, are a valuable tool to investigate the role of the actomyosin machinery in podosome mesoscale dynamics. Finally, we are grateful for the suggestion of exploiting more the scratched surfaces and hope the reviewer will agree that our new experiments satisfactorily establish the relationship between the mesoscale dynamics and mechanosensing.

Specific comments:

3) <Concerning Fig7d and g in page 26: it is clearly shown that the velocity magnitude of LifeAct-RFP labeled actin network is substantially reduced after CytD treatment in the representative of the flow magnitude in Fig7b and paired comparison of quantification of flow magnitude in cells before and after adding CytD. However, the histogram of flow

velocity of LifeAct-RFP after treated with CytD is virtually the same as that before CytD treatment. This seems surprising and should be explained better.>

The apparent discrepancy between panel 7d and panel 7g is due to the fact that all the ROIs are used to calculate the average velocity of panel 7d, thus including those without a vector that are considered to have velocity 0. In panel 7f and 7g, we look in more details at the ROIs to understand where the lower average velocity comes from after CytoD treatment. We find that the lower velocity is due to a higher number of ROIs without a vector (i.e.: with velocity 0) after CytoD treatment with respect to untreated cells (as depicted in 7f). Panel 7g only shows the velocity values of all the ROIs that have a vector and indeed these values appear to be similar before and after CytoD treatment. Although we did explain this in the original manuscript in the M&M section '*Vector velocity magnitude quantification*', the reviewer's comment prompted us to explain this more clearly in the revised main text. We therefore added an explanatory sentence in the revised manuscript (see **page 15**).

4) <Minor comments:

Line 309: Fig. 8g should be Fig. 7g>

Thank you for pointing this out, we have corrected the text.

Reviewer #2 (Remarks to the Author)

The authors have addressed majority of the concerns raised by the reviewer and have clarified the text and improved the statistics/reproducibility of the findings. The authors have included a new paragraph demonstrating that the topographical cues promote the directionality of podosome cluster movement and scratched surfaces represent nucleation sites for podosomes reformation. These findings are interesting and give additional value to the manuscript.

Concerning the role of myosin IIA in the podosome cluster mesoscale dynamics, the reviewer fully understands the difficulties working with primary cells. However, the SIM images illustrating the effect of ML7+Y27632 included in the new Suppl Fig 8c could be improved. It is difficult to assess the effect of ML7+Y27632 from these images, the zoomed area in Fig 8a is not comparable to the zoom in Suppl. Fig 8c. Details cannot be seen (actin filaments radiating from the cores). Please highlight the individual podosomes and demonstrate the presence/absence of Myosin IIA in the actin filaments radiating from the cores.

Fig 7: In all the figure, CytoD is written 2 different ways: CytD or CytoD. The same applies also to the main text (materials and method section). Fig 7 legend: a) 2.5 µg/ml CytoD is not correctly written. "Pooling all the cells used for panels d, e, f and g", should be "Pooling all the cells used for panels d, e, f and h"

In Fig 7, it is not clear in panels d,e,f and h why in the legend it states: Each dot represents a single cell and line shows the mean (page 27, line 1001). In these graphs the lines connect the before and after points corresponding to the same cell. Please clarify.

The same applies to legend Fig 7 c and d (page 8, line 183). Please clarify.

Suppl. Fig 8 legend is messy/incorrect and the anti-vinculin antibody should be anti-Myosin IIA antibody.

Reviewer #3 (Remarks to the Author)

In this revision, the authors have addressed all of the scientific and technical concerns that I have raised for the original submission, and thus the revision is much approved. My recommendation to the editor is to proceed as appropriate.

Reviewer #4 (Remarks to the Author)

Overall, Meddens et al. convincingly addressed my comments but have still not correctly motivated their choice of LifeAct as discussed in <Point 1>. LifeAct is indeed a versatile fluorescent marker which has been demonstrated many times to accurately visualize filamentous actin in cells (see for example the publications pointed out by the authors; in particular Riedl et al, Nature Communications 2008). The authors also emphasize correctly that inappropriate fluorescent labelling or vast over-expression of LifeAct-GFP or Actin-GFP can lead to artifacts.

However, I feel that the author have not quite addressed the potential issue with LifeAct reporting on

actin dynamics. LifeAct continuously associates/dissociates to/from actin filaments undergoing constant reaction-diffusion dynamics; diffusing in the cytoplasm and attaching to the tip or bulk of actin filaments and then eventually detaching and diffusing away. In contrast actin monomers diffuse in the cytoplasm and then specifically bind to the barbed ends and eventually detach after a finite time from the pointed end of the actin filaments. If the actin filaments are short and turnover acts at the same characteristic time-scale as the binding of LifeAct molecules as it is the case with short filaments; the dynamics are not correctly reflected. Consequently, at scenarios where actin polymerization acts on the same time-scale as LifeAct binding, such as during the inhibition of actin polymerization by cytochalasin D, the characteristic velocity profiles can be

qualitatively and quantitatively be different compared to those reported by molecular actin dynamics. Cortical actin structures and actin in the lamellipodium comprise two F-actin populations: very short Arp2/3-mediated F-actin and long formin-mediated F-actin (Bovellan et al, Current Biology 2014; Fritzsche et al, MBoC 2013). The polymerization rate depends on the concentration of free actin monomers and differs from cell to cell and depend on the physiological conditions of the cell. Hence, it is not accurate to refer to different cell types having measured the speed of actin polymerization.

The authors argue that the diffusion of LifeAct-GFP is faster than those of actin monomers but presumably mean that they actually measured the turnover dynamics of both LifeAct-GFP and Actin-GFP molecules in the podosomes, reporting on the reaction-diffusion dynamics. The diffusion of actin monomers and LifeActin molecules is comparable in the cytoplasm of cells ($20\text{-}25\mu\text{m}^2/\text{s}$); considering the size of Actin-GFP (70kDa) and LifeAct-GFP molecules (30kDa). Specifically, LifeAct binding is independent on the location of the filament binding and thus much faster than actin turnover. Hence, LifeAct is only an appropriate marker to report on actin flow when the turnover time of LifeAct is significantly faster than the turnover time of actin monomers for both actin filament subpopulations. The turnover time of Arp2/3-mediated and formin-mediated F-actin has been estimated to 1s and 10s, respectively (Fritzsche et al, MBoC 2013). LifeAct turnover shall therefore be $\ll 1\text{s}$.

I would like to ask the authors to address this better in the manuscript and/or eventually clarify the potential issues with the velocity distributions on short time-scales in the discussion section. The authors may want to add the FRAP curves to the Supplementary Materials. This should include all wording of 'actin polymerization' to actin flow; e.g. Fig. 7 caption. The authors may want to quantify the LifeAct turnover time and compare those with the characteristic time-scale of their measured flow rates in order to be prepared for future criticism.

Response to reviewer's comments

Reviewer #2 (Remarks to the Author):

The authors have addressed majority of the concerns raised by the reviewer and have clarified the text and improved the statistics/reproducibility of the findings. The authors have included a new paragraph demonstrating that the topographical cues promote the directionality of podosome cluster movement and scratched surfaces represent nucleation sites for podosomes reformation. These findings are interesting and give additional value to the manuscript.

Concerning the role of myosin IIA in the podosome cluster mesoscale dynamics, the reviewer fully understands the difficulties working with primary cells. However, the SIM images illustrating the effect of ML7+Y27632 included in the new Suppl Fig 8c could be improved. It is difficult to assess the effect of ML7+Y27632 from these images, the zoomed area in Fig 8a is not comparable to the zoom in Suppl. Fig 8c. Details cannot be seen (actin filaments radiating from the cores). Please highlight the individual podosomes and demonstrate the presence/absence of Myosin IIA in the actin filaments radiating from the cores.

We updated Supplementary Fig. 8 to better show individual podosomes. Supplementary Fig. 8 now shows similar zoomed in images as Fig 8a.

Fig 7: In all the figure, CytoD is written 2 different ways: CytD or CytoD. The same applies also to the main text (materials and method section). Fig 7 legend: a) 2.5 µg/ml CytoD is not correctly written.

We changed all incidences to CytoD throughout the manuscripts, legends and within Fig. 7.

"Pooling all the cells used for panels d, e, f and g", should be "Pooling all the cells used for panels d, e, f and h"

We corrected the legend.

In Fig 7, it is not clear in panels d,e,f and h why in the legend it states: Each dot represents a single cell and line shows the mean (page 27, line 1001). In these graphs the lines connect the before and after points corresponding to the same cell. Please clarify.

The same applies to legend Fig 7 c and d (page 8, line 183). Please clarify.

Thank you for pointing out this mistake. These lines connect the same cells before and after treatment and the legends are corrected accordingly.

Suppl. Fig 8 legend is messy/incorrect and the anti-vinculin antibody should be anti-Myosin IIA antibody.

The mistake was corrected and the legend was made clearer

Reviewer #3 (Remarks to the Author):

In this revision, the authors have addressed all of the scientific and technical concerns that I have raised for the original submission, and thus the revision is much approved. My recommendation to the editor is to proceed as appropriate.

Reviewer #4 (Remarks to the Author):

Overall, Meddens et al. convincingly addressed my comments but have still not correctly motivated their choice of

LifeAct as discussed in . LifeAct is indeed a versatile fluorescent marker which has been demonstrated many times to accurately visualize filamentous actin in cells (see for example the publications pointed out by the authors; in particular Riedl et al, Nature Communications 2008). The authors also emphasize correctly that inappropriate fluorescent labelling or vast over-expression of LifeAct-GFP or Actin-GFP can lead to artifacts.

However, I feel that the author have not quite addressed the potential issue with LifeAct reporting on actin dynamics. LifeAct continuously associates/dissociates to/from actin filaments undergoing constant reaction-diffusion dynamics; diffusing in the cytoplasm and attaching to the tip or bulk of actin filaments and then eventually detaching and diffusing away. In contrast actin monomers diffuse in the cytoplasm and then specifically bind to the barbed ends and eventually detach after a finite time from the pointed end of the actin filaments. If the actin filaments are short and turnover acts at the same characteristic time-scale as the binding of LifeAct molecules as it is the case with short filaments; the dynamics are not correctly reflected. Consequently, at scenarios where actin polymerization acts on the same time-scale as LifeAct binding, such as during the inhibition of actin polymerization by cytochalasin D, the characteristic velocity profiles can be qualitatively and quantitatively be different compared to those reported by molecular actin dynamics. Cortical actin structures and actin in the lamellipodium comprise to F-actin populations: very short Arp2/3-mediated F-actin and long formin-mediated F-actin (Bovellan et al, Current Biology 2014; Fritzsche et al, MBoC 2013). The polymerization rate depends on the concentration of free actin monomers and differs from cell to cell and depend on the physiological conditions of the cell. Hence, it is not accurate to refer to different cell types having measured the speed of actin polymerization.

The author argue that the diffusion of LifeAct-GFP is faster than those of actin monomers but presumably mean that they actually measured the turnover dynamics of both LifeAct-GFP and Actin-GFP molecules in the podosomes, reporting on the reaction-diffusion dynamics.

The reviewer is right, we indeed measured reaction-diffusion kinetics, not diffusion. We should have been more precise in our wording in the previous response.

The diffusion of actin monomers and LifeActin molecules is comparable in the cytoplasm of cells ($20\text{-}25\mu\text{m}^2/\text{s}$); considering the size of Actin-GFP (70kDa) and LifeAct-GFP molecules (30kDa). Specifically, LifeAct binding is independent on the location of the filament binding and thus much faster than actin turnover. Hence, LifeAct is only an appropriate marker to report on actin flow when the turnover time of LifeAct is significantly faster than the turnover time of actin monomers for both actin filament subpopulations.

The turnover time of Arp2/3-mediated and formin-mediated F-actin has been estimated to 1s and 10s, respectively (Fritzsche et al, MBoC 2013). LifeAct turnover shall therefore be $\ll 1\text{s}$.

I would like to ask the authors to address this better in the manuscript and/or eventually clarify the potential issues with the velocity distributions on short time-scales in the discussion section.

We agree with the reviewer that LifeAct is not an appropriate tool to report F-actin dynamics of very fast processes, or we should say of processes that are as fast as the turnover time of LifeAct. However, as already explained in the response to the reviewer and mentioned in the manuscript (see page XX), our imaging rate for all the STICS data is 15 sec frame^{-1} , and the vector velocities are calculated over 10 frames, meaning that the turnover of LifeAct binding to actin is too fast to be picked up by STICS, hence it does not influence our measurements. To make this concept even clearer, we now added a short sentence in the manuscript (see page 12) where we explain that care should be taken if STICS is going to be performed using much faster imaging rates. Therefore, with the STICS as applied in this study, we are specifically detecting relatively 'slow' processes. We believe our additional comment in the manuscript are really sufficiently clear for the readership.

The authors may want to add the FRAP curves to the Supplementary Materials. This should include all wording of 'actin polymerization' to actin flow; e.g. Fig. 7 caption. The authors may want to quantify the LifeAct turnover time and compare those with the characteristic time-scale of their measured flow rates in order to be prepared for future criticism.

We have added the FRAP curves to Supplementary Figure 3 and extended and clarified the results section where we compare LifeAct and actin. Furthermore, as also answered for the previous comment, we now discuss the limitations of LifeAct as a probe for fast F-actin dynamics in the discussion (see page 12).